EMBO
Molecular Medicine

# Activation of the epithelial sodium channel (ENaC) leads to cytokine profile shift to pro-inflammatory in labor

Xiao Sun[1,†], Jing Hui Guo[1,2,3,†], Dan Zhang[4], Jun-jiang Chen[1,2,3], Wei Yin Lin[1], Yun Huang[4], Hui Chen[1], Wen Qing Huang[1], Yifeng Liu[4], Lai Ling Tsang[1], Mei Kuen Yu[1,2], Yiu Wa Chung[1], Xiaohua Jiang[1], Hefeng Huang[4,5], Hsiao Chang Chan[1,*] (iD) & Ye Chun Ruan[2,**] (iD)

## Abstract

The shift of cytokine profile from anti- to pro-inflammatory is the most recognizable sign of labor, although the underlying mechanism remains elusive. Here, we report that the epithelial sodium channel (ENaC) is upregulated and activated in the uterus at labor in mice. Mechanical activation of ENaC results in phosphorylation of CREB and upregulation of pro-inflammatory cytokines as well as COX-2/PGE[2] in uterine epithelial cells. ENaC expression is also upregulated in mice with RU486-induced preterm labor as well as in women with preterm labor. Interference with ENaC attenuates mechanically stimulated uterine contractions and significantly delays the RU486-induced preterm labor in mice. Analysis of a human transcriptome database for maternal–fetus tissue/blood collected at onset of human term and preterm births reveals significant and positive correlation of ENaC with labor-associated pro-inflammatory factors in labored birth groups (both term and preterm), but not in non-labored birth groups. Taken together, the present finding reveals a pro-inflammatory role of ENaC in labor at term and preterm, suggesting it as a potential target for the prevention and treatment of preterm labor.

**Keywords** ENaC; labor/parturition; preterm labor; pro-inflammatory
**Subject Categories** Immunology; Urogenital System

## Introduction

Labor or parturition, the end step of pregnancy, remains poorly understood (Muglia & Katz, 2010; Rubens *et al*, 2014), which

accounts for the lack of effective method to prevent or predict preterm labor, a leading cause of neonatal death and disability (Liu *et al*, 2012; Iams, 2014b; Romero *et al*, 2014). Given the similar clinical events involved, both term and preterm labor are considered to share common pathways of labor (Romero *et al*, 2014). The shift in cytokine profile from anti-inflammatory to pro-inflammatory and uterine activity from quiescent to contractile is the most recognizable sign of labor (Simhan & Caritis, 2007; Renthal *et al*, 2010, 2013; Tan *et al*, 2012; Adams Waldorf & McAdams, 2013; Romero *et al*, 2014). However, the molecular mechanism underlying the "shift" of these events remains elusive.

Among the recognized maternal or fetal derived signals associated with the initiation of labor, mechanical forces generated by either the growing fetus or initial uterine contractions are believed to be important (Lye *et al*, 2001). Mechanical stimulations are observed to induce uterine production of pro-inflammatory cytokines (e.g., IL-8; Maehara *et al*, 1996; Loudon *et al*, 2004; Shynlova *et al*, 2008) and prostaglandins (PGs; Challis *et al*, 2000; Sooranna *et al*, 2004), which in turn potently evoke uterine contractions (Wray, 1993) and therefore may underlie a positive-feedback loop resulting in increasingly powerful expelling forces required for labor (Wray, 1993; Lye *et al*, 2001). A large number of studies are dedicated to myometrium muscle cells and suggest their essential involvement in the positive-feedback loop at labor (Wray, 1993; Lye *et al*, 2001). Interestingly, epithelial cells lining the uterine cavity as part of the decidua are also believed to be a source of PGs at term (Dong *et al*, 1996; Olson, 2003; Satoh *et al*, 2013), although whether and how these epithelial cells play a role in the process of labor is not well studied.

We have previously demonstrated that ENaC in the endometrial epithelium plays an essential role in embryo implantation by activating $Ca^{2+}$/cAMP response element binding protein (CREB) and

1  Epithelial Cell Biology Research Centre, School of Biomedical Sciences, Faculty of Medicine, The Chinese University of Hong Kong, Hong Kong, China
2  Department of Biomedical Engineering, Faculty of Engineering, The Hong Kong Polytechnic University, Hong Kong, China
3  Department of Physiology, School of Medicine, Jinan University, Guangzhou, China
4  Department of Reproductive Endocrinology, Women's Hospital, School of Medicine, Zhejiang University, Hangzhou, China
5  International Peace Maternal and Child Health Hospital, Shanghai Jiao Tong University, Shanghai, China
   *Corresponding author. Tel: +852 39436839; E-mail: hsiaocchan@cuhk.edu.hk
   **Corresponding author. Tel: +852 34008084; E-mail: sharon.yc.ruan@polyu.edu.hk
   †These authors contributed equally to this work

cyclooxygenase 2 (COX-2) to produce prostaglandins E 2 (PGE$_2$), which acts as a paracrine factor for stromal cell decidualization, the prerequisite of embryo implantation (Ruan et al, 2012). Interestingly, many of the labor-associated inflammatory cytokines, such as IL-6, IL-8, and TNFα, are known to be downstream targets of either CREB or PGE$_2$ (Wen et al, 2010; Srivastava et al, 2012). Moreover, the open probability of ENaC has been shown to be increased by mechanical stimuli (Fronius & Clauss, 2008; Shi et al, 2013), although the physiological significance of this mechano-sensitivity has not been well demonstrated. The mechano-sensitivity of ENaC and its ability to regulate the CREB/COX-2/PGE$_2$ axis led us to hypothesize that ENaC might play a role in mechano-sensing and signal transduction leading to the shift in the cytokine profile and uterine contractile state during labor process, at term or preterm. We undertook the present study to test this hypothesis and explore treatment strategy for preterm labor.

## Results

### ENaC is upregulated and activated at labor in mice

To establish a possible role of ENaC in labor, we first examined the expression level of ENaC in the uterus during late gestation approaching parturition in mice. ENaCα, the rate-limiting subunit of ENaC, showed a gradual increase at the mRNA level from 15 d.p.c. to the labor day (19 d.p.c.) in fetus/placenta-removed uterine tissues (Fig 1A). Western blot analysis showed that ENaCα and its cleaved form, indicative of ENaC activation (Kleyman et al, 2009), were substantially increased at 18 and 19 d.p.c. as compared to 15 d.p.c. (Fig 1B). Immuno-staining for ENaCα in mouse uterus also showed stronger signal intensity detected at 19 d.p.c. compared to that at 15 d.p.c. in decidua epithelial cells (higher level detected at the apical membrane) lining the uterine cavity and placental regions (Fig 1C), suggesting a potential role of ENaC in the process of labor.

### Mechano-activation of ENaC-dependent signaling and pro-inflammatory cytokines in the uterine epithelial cells

We had previously shown that ENaC activation could lead to phosphorylation/activation of CREB (Ruan et al, 2012). CREB is a key transcription factor known to promote the expression of pro-inflammatory cytokines (Wen et al, 2010). Given the known mechano-sensitivity of ENaC (Shi et al, 2013), we next investigated whether ENaC-dependent CREB activation and subsequent upregulation of pro-inflammatory cytokines could be elicited mechanically during labor. We adopted ex vivo stretch stimulation to uterine preparations collected from mice at 19 d.p.c. (See Materials and Methods). After 1-h stretch, immunofluorescence labeling for phosphorylated CREB (pCREB) in the uterine tissues showed abundant pCREB expression in the nuclei of the uterine epithelial cells (Fig 2A), whereas other cell types (e.g., muscle cells) showed much weaker or absence of fluorescence. Quantification of the fluorescence intensity revealed that the epithelial nuclear pCREB expression was significantly stronger in the stretched uterine preparations as compared to that in non-stretched ones (Fig 2B). Moreover, the stretch-induced nuclear expression of pCREB was abolished in the presence of amiloride (10 μM, Fig 2A and B), a selective blocker of

ENaC, indicating ENaC-dependent epithelial CREB activation by mechanical stimulations. This was also tested in a human endometrial epithelial cell line, Ishikawa (ISK) cells. Results showed that stretch (15% elongation) of the cells for 30 min increased phosphorylation of CREB, which was blocked by pretreatment with amiloride (10 μM, Fig EV1).

We further investigated whether the stretch-induced ENaC/CREB activation could lead to a shift in cytokine profile, switching on the pro-inflammatory cytokines involved in labor. Indeed, the stretch of mouse uterine tissues upregulated IL-6 and TNFα, which were both significantly inhibited by amiloride (10 μM, Fig 2C), suggesting the involvement of ENaC in the upregulation of pro-inflammatory cytokines. To confirm this, we next used ISK cells in conjunction with ENaC knockdown by shRNAs. To better mimic mechanical stimuli in the uterus during labor, which are mostly rhythmic contractile forces, we applied a cyclic stimulation of stretch (15% elongation) at frequency of 1 Hz as previously reported (Korita et al, 2002; Yoshida et al, 2002) to ISK cells grown on flexible culture supports. shRNAs against ENaCα (shENaCα), the rate-limiting subunit of ENaC, were used to knockdown ENaC. Quantitative PCR showed ~ 50% reduction in ENaCα mRNA level in ISK cells treated with the shENaCα as compared to the cells treated with non-silencing control shRNAs (shNC; Fig 2D). The cyclic stretch-induced increases in mRNA levels of IL-6, IL-8, TNFα, and COX-2 were all significantly inhibited by ENaC knockdown (Fig 2E), confirming the involvement of ENaC. These results suggest that the stretch-induced upregulation of pro-inflammatory cytokines may be a subsequence to the mechano-activation of ENaC.

To demonstrate mechano-activation of ENaC directly, we also examined ENaC current activation in response to varied mechanical force in the human endometrial epithelial cells using an automatic patch-clamp system (See Materials and Methods). Negative pressures (40, 80, and 120 mPa) were used to stretch the cells during whole-cell recording. At holding voltage of −80 mV, an inward whole-cell current was observed as the stretching force was increased to 120 mPa, which was subsequently abolished by amiloride (10–20 μM, Fig EV2), indicating stretch-induced activation of ENaC.

Since PGE$_2$ is implicated in the positive-feedback loop of uterine contractions during labor process, we also examined COX-2 level in the stretched uterine tissues and the PGE$_2$ concentration in the incubating media. The results showed that the levels of both COX-2 (Fig 2F) and PGE$_2$ (Fig 2G) were significantly elevated by overnight uterine stretch, which were abrogated by the treatment with amiloride (10–50 μM, Fig 2F and G). Of interest, overnight stretch also induced upregulation of all three ENaC subunits, as compared to non-stretched ones (Fig 2H). The cleaved/activated form of ENaCα was also enhanced after the stretch (Fig 2H), suggesting activation of ENaC by mechanical forces. Taken together, these results suggest the involvement of ENaC in mechanically stimulated upregulation of the pro-inflammatory cytokines/factors.

### Epithelial cells and ENaC are involved in stretch-facilitated uterine contractions

The suggested role of ENaC in mechanically stimulated increases in COX-2 and PGE$_2$, a potent elicitor of uterine contractions, in epithelial cells, prompted us to examine possible involvement of uterine epithelium and ENaC in regulating uterine contractions in response

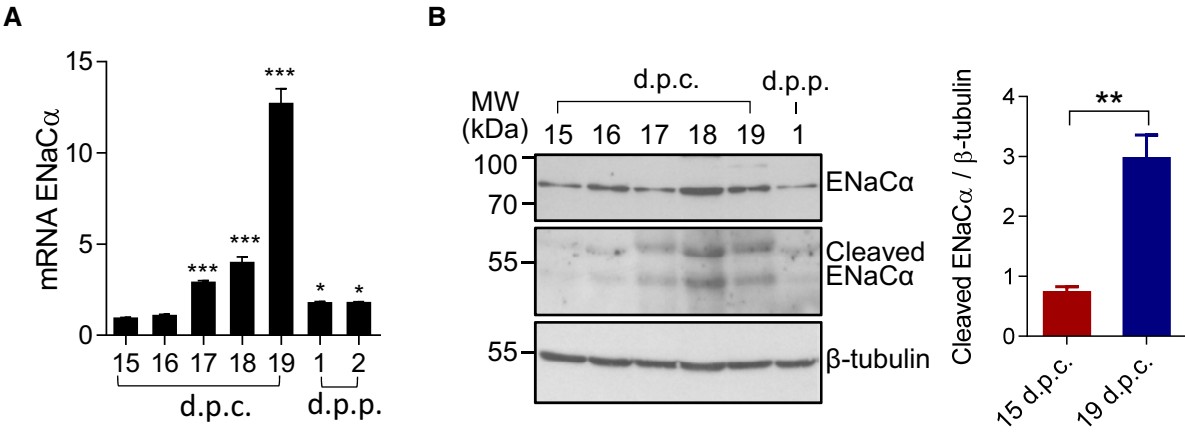

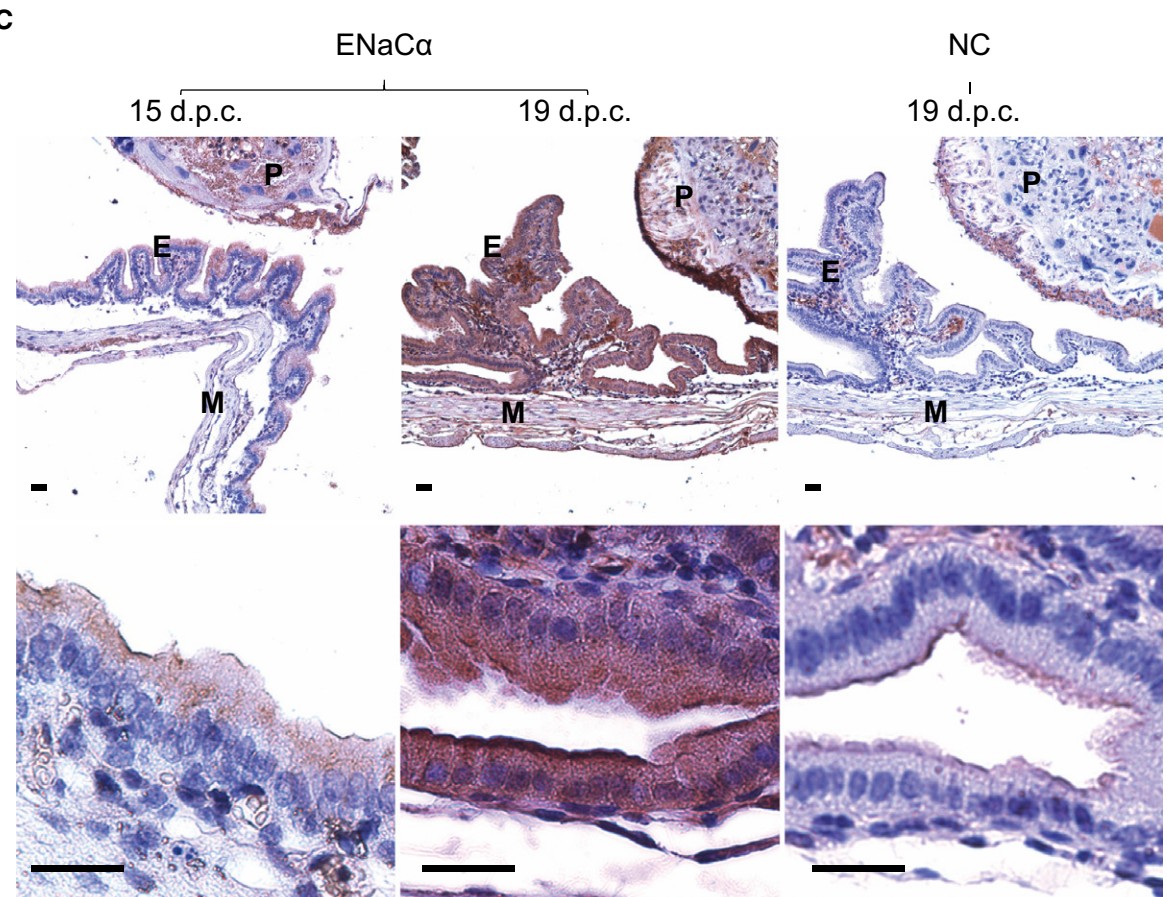

**Figure 1. ENaC is upregulated at labor in mice.**

A   Quantitative PCR (qPCR) analysis of ENaCα in fetus/placenta-removed uterine tissues from pregnant mice at 15–19 days post-coitum (d.p.c.) and 1–2 day(s) post-partum (d.p.p.). Data are shown as mean ± SEM. $n = 4$. *$P < 0.05$, ***$P < 0.001$, one-way ANOVA with Tukey's multiple comparisons test, compared to 15 d.p.c. 18S rRNA was used as a housekeeping gene for normalization.

B   Western blot (with quantification shown on the right) analysis of ENaCα in fetus/placenta-removed uterine tissues from pregnant mice at 15–19 d.p.c. and 1 d.p.p. Data are shown as mean ± SEM. $n = 4$. Two-tailed unpaired Student's $t$-test with unequal variance corrected by Welch's correction. **$P < 0.01$. β-tubulin was blotted as a loading control.

C   Immuno-histochemical labeling for ENaCα in uterine tissues from mice at 15 or 19 d.p.c. E: epithelial tissues, P: placenta, and M: myometrium. NC: negative control done with the absence of primary antibody. Nuclei are labeled with hematoxylin. Scale bar, 20 μm.

Data information: Exact $P$-values are listed in Appendix Table S1.
Source data are available online for this figure.

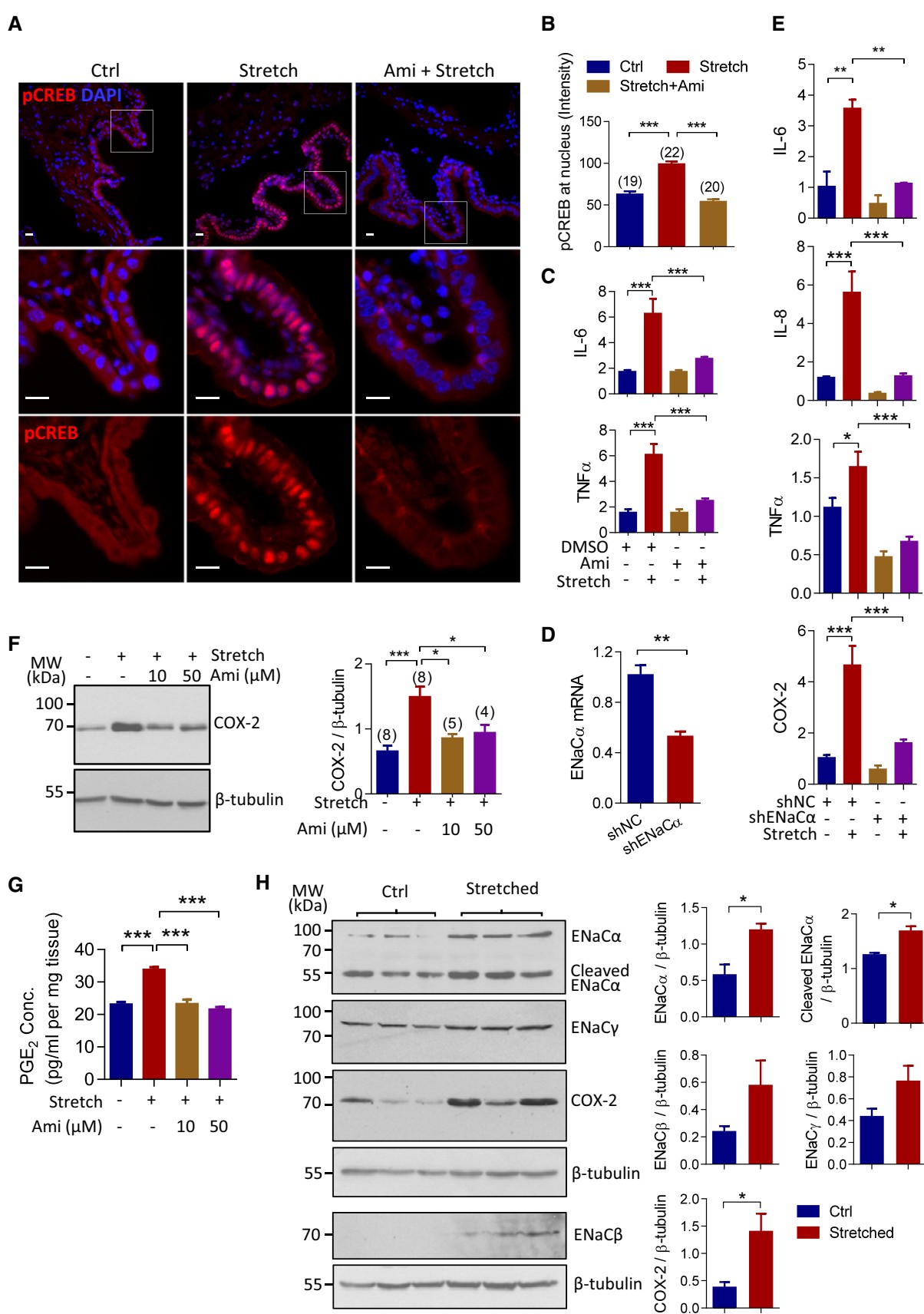

**Figure 2.**

**Figure 2.  Mechano-activation of ENaC-dependent signaling and pro-inflammatory cytokines in uterine epithelial cells.**

A, B   Immunofluorescence labeling for phosphorylated CREB (pCREB, A) with quantification of fluorescence intensity in nuclei of epithelial cells (B) in mouse uterine tissues (19 d.p.c.), either kept slack (Ctrl) or stretched and incubated for 1 h (see Materials and Methods), in the presence or absence of amiloride (Ami, 10 μM), a selective blocker of ENaC. Data are shown as mean ± SEM. ***$P < 0.001$, one-way ANOVA with Tukey's multiple comparisons test, $n$ is shown in each column. Scale bars, 10 μm.

C      qPCR analysis of IL-6 and TNFα mRNA levels in mouse uterine tissues with or without 1-h stretch in the absence or presence of Ami (10 μM). Data are shown as mean ± SEM. $n = 16$. ***$P < 0.001$, one-way ANOVA with Tukey's multiple comparisons test. Data are relative mRNA levels normalized to GAPDH levels.

D      qPCR of ENaCα in ISK cells transfected with shRNAs against ENaCα (shENaCα) or non-silencing control (shNC). Data are shown as mean ± SEM. **$P < 0.01$, two-tailed unpaired Student's $t$-test, $n = 4$. Data are relative mRNA levels normalized to GAPDH levels.

E      qPCR of IL-6, IL-8, TNFα, and COX-2 in ISK cells transfected with shENaC or shNC with (+) or without (−) 1-h stretch. Data are shown as mean ± SEM. *$P < 0.05$, **$P < 0.01$, ***$P < 0.001$, one-way ANOVA with Tukey's multiple comparisons test, $n = 3$ (IL-6), 6 (IL-8), or 4 (TNFα and COX-2). Data are relative mRNA levels normalized to GAPDH levels.

F, G   Western blotting with quantification for COX-2 in mouse uterine preparations (F) and ELISA analysis of $PGE_2$ levels in uterine preparation-incubated media (G) with (+) or without (−) overnight stretch, in the absence (−) or presence (+) of Ami (10–50 μM). Data are shown as mean ± SEM. *$P < 0.05$, ***$P < 0.001$, one-way ANOVA with Tukey's multiple comparisons test, $n$ is shown in each column in (F) and $n = 3$ in (G).

H      Western blotting with quantification for ENaCα, β, γ, and COX-2 expression in mouse uterine preparations (19 d.p.c.), either kept slack (Ctrl) or stretched and incubated overnight. β-tubulin was blotted as a loading control. Data are shown as mean ± SEM. $n = 3$. *$P < 0.05$, two-tailed unpaired Student's $t$-test.

Data information: Exact $P$-values are listed in Appendix Table S1.
Source data are available online for this figure.

to mechanical stimulation. The contractility of mouse uterine tissue preparations (19 d.p.c.) was measured *ex vivo*, which exhibited rhythmic spontaneous contractions. Stretches by 10–40% elongation of *in situ* length ($L_0$) of the preparations were applied at the end of a rhythmic spontaneous contractive phase, which substantially advanced the following contractive phase (Fig 3A). The stretch-induced phase-advancing effect was calculated, according to a previously reported method (Kasai *et al*, 1995), as $(T_0 - T_1)/(T_0 - T_s)$ (See Materials and Methods). As shown in Fig 2B, the maximal phase-advancing effect on 19 d.p.c. uterine preparations was achieved by 20% $L_0$ stretch, which was then used in the following experiments.

To explore a possible role of the epithelium, we denuded the epithelium from the uterine preparations (Fig 3C) and this resulted in significant inhibition of the stretch (20% $L_0$)-induced phase advancement as compared to the intact tissues (Fig 3D), suggesting a role of the epithelium in regulating uterine contractions. Similarly, inhibition of ENaC by amiloride (10–200 μM), starting from a low dose (10 μM), significantly inhibited the stretch (20% $L_0$)-induced phase advancement in a concentration-dependent manner (Fig 3E), indicating the involvement of ENaC in stretch-facilitated contractions. Of note, in contrast to the time of parturition (19 d.p.c.), the stretch (20% $L_0$)-induced phase-advancing effect was found significantly less in uterine preparations from mice at 15 d.p.c. (Fig 3F). This is consistent with the time course expression profile of ENaC (Fig 1), with stronger uterine contractions observed when ENaC expression level is maximal, further suggesting the involvement of ENaC in regulating uterine contractions during labor process.

**Interference with ENaC delays RU486-induced preterm labor in mice**

We further explored the role of ENaC in preterm labor and possible prevention in two well-established preterm labor mouse models, induced by RU486 (a progesterone receptor antagonist; Dudley *et al*, 1996) and lipopolysaccharide (LPS; Gross *et al*, 2000), respectively. We subcutaneously injected RU486 (200 μg per mouse) into pregnant mice at 15 d.p.c., which resulted in preterm labor in ~ 20 h. Uterine tissues were collected either at preterm labor from RU486-treated mice or time-matched vehicle-injected ones as

control. RU486-treated uterus showed significantly higher protein levels of ENaCα and its cleaved/activated form, as compared that in the control group (Fig 4A).

If the upregulation of ENaC by RU486 is involved in preterm labor in these mice, blocking ENaC would prevent or delay RU486-induced labor. To test this, amiloride (initial dose, 10 mg/kg body weight) was intraperitoneally injected into the mice right before and every 7–8 h after the RU486 injection (200 μg per mouse). Up to four injections of amiloride were done before the labor was initiated and the cumulative doses of amiloride were used from 10 to 40 mg/kg body weight. Results showed that all the RU486-treated control mice ($n = 26$) gave birth within 24 h; however, amiloride dose-dependently reduced the number of RU486-induced birth, with none of the mice ($n = 9$) delivered within 24 h at the cumulative dose of 40 mg/kg (Fig 4B). We also treated the mice with atosiban (initial dose, 10 mg/kg body weight), an antagonist of oxytocin receptor currently used for preventing preterm labor in humans (Worldwide Atosiban versus Beta-agonists Study Group, 2001; Saleh *et al*, 2013) in the same fashion. Almost all of the atosiban-treated mice delivered before the 4[th] injection, and thus, its cumulative dose was used up to 30 mg/kg body weight, at which, 85.7% (6 out of 7) of the atosiban-treated mice delivered within 24 h, whereas only 40% (2 out of 5) were observed in mice treated with the same dose (30 mg/kg cumulatively) of amiloride (Fig 4B). We also injected siRNAs against ENaCα (siENaCα, 600 pmole per mouse, i.p.) into pregnant mice at 14 and 15 d.p.c. before RU486 injection (200 μg per mouse). The siENaCα treatment resulted in ~ 40% lower ENaCα expression in the uterus, as compared to mice treated with control siRNAs (siNC). Even at this low transfection efficiency, knockdown of ENaC significantly reduced the number of births delivered in 20 h after RU486 injection (33.3%, 2 out of 6) in siENaCα-treated mice, whereas 100% (5 out of 5) in the control siNC-treated group (Fig 4C).

In another preterm mouse model by injection of LPS mimicking infection-associated preterm labor, overwhelming inflammation as indicated by extensive redness of intraperitoneal tissues was observed. The expression levels of ENaCα and γ expression in the uterus at labor were found to be decreased even though COX-2 expression was increased (Fig EV3A) in LPS-treated mice as compared to the controls, suggesting that upregulation of COX-2 in

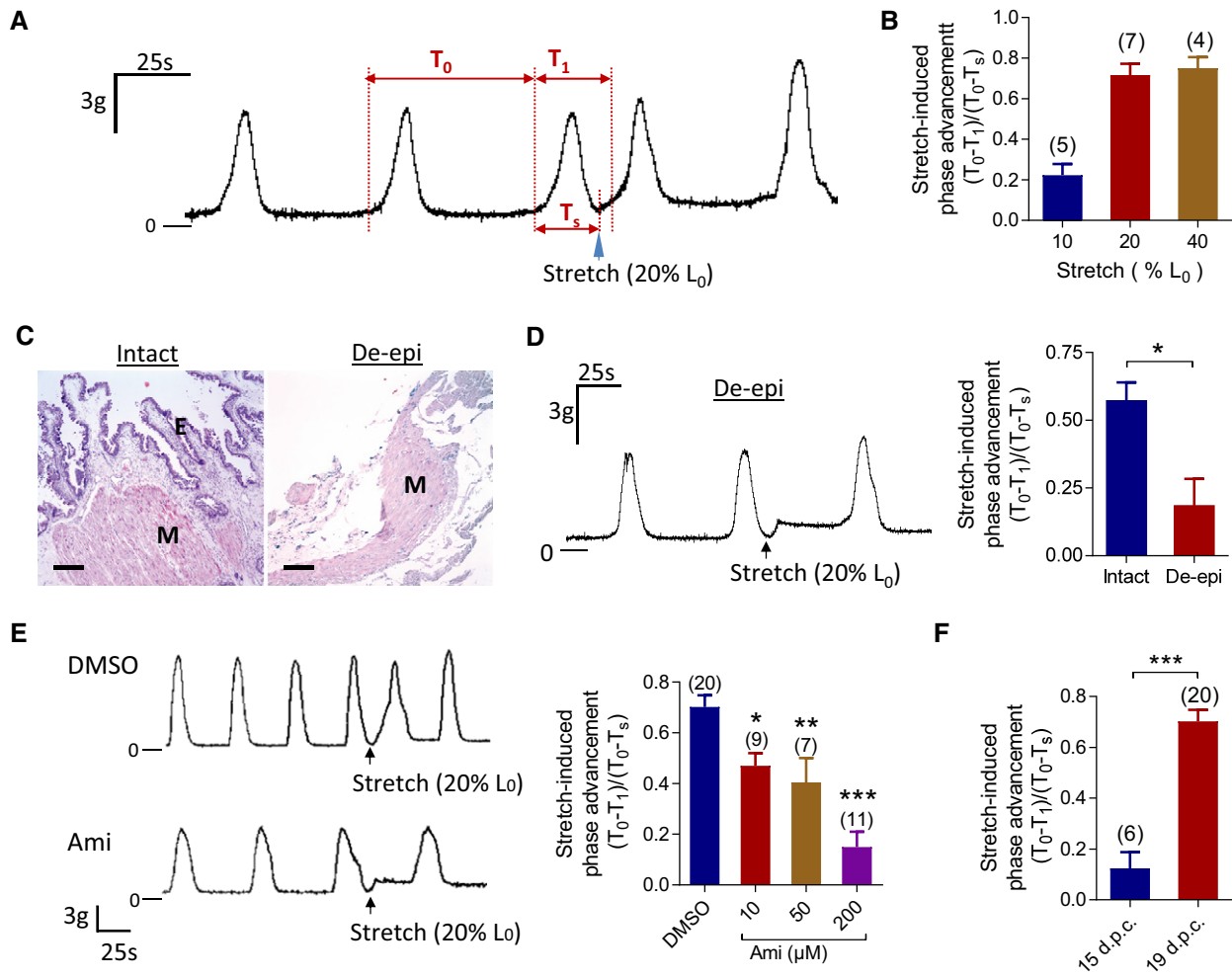

**Figure 3. Epithelial cells and ENaC are involved in mechano-facilitated uterine contractions in mice.**

A, B  Representative mechanogram (A) with quantification (B) showing stretch-induced advancement of spontaneous rhythmic contractile phase in uterine preparations collected from mice at 19 d.p.c. Stretch varied by 10–40% elongation of original uterine length, $L_0$. The advancing effect is calculated by $(T_0-T_1)/(T_0-T_s)$, where $T_0$ is the time between onsets of two consecutive contractions prior to the stretch; $T_1$, period between two consecutive contractile onsets before and after the stretch; and $T_s$, the time when the stretch is initiated as measured from the onset of the previous contraction. Data are shown as mean ± SEM. $n$ is shown in each column.
C  H&E staining of uterine tissues with intact or denudated (de-epi) epithelium. E: epithelium. M: myometrium. Scale bars, 100 μm.
D  Representative mechanogram with quantification showing the effect of removal of the epithelium (de-epi) on stretch (20% $L_0$)-facilitated uterine contractions. Data are shown as mean ± SEM. *$P < 0.05$, two-tailed paired Student's $t$-test. $n = 5$.
E  Representative mechanograms (left) of uterine contractions before and after the stretch (20% $L_0$) are applied in the absence (DMSO) or presence of amiloride (Ami, 200 μM) with quantification (right) showing the effect of Ami (10–200 μM) on stretch-facilitated uterine contractions. Data are shown as mean ± SEM. $n$ is shown in each column. *$P < 0.05$; **$P < 0.01$; ***$P < 0.001$ compared to the group treated with DMSO as the control, one-way ANOVA with Tukey's multiple comparisons test.
F  Stretch-induced contractile phase-advancing effect in mouse uterus at 15 or 19 d.p.c. Data are shown as mean ± SEM. ***$P < 0.001$, two-tailed unpaired Student's $t$-test. $n$ is shown in each column.

Data information: Exact $P$-values are listed in Appendix Table S1.

the infection-associated preterm labor may be ENaC-independent. As expected, the treatment with amiloride failed to delay LPS-induced preterm labor in mice (Fig EV3B).

### ENaC expression is upregulated in spontaneous preterm labor women

We next tested whether ENaC is involved in term or preterm labor in humans. Since ENaC was observed to be upregulated in RU486-induced preterm labor mice (Fig 4A), we speculated that abnormal upregulation in ENaC expression might occur in preterm labor in humans. Due to the ethical concerns and limitation of obtaining uterine samples, and given that placental tissues also showed increased ENaC expression at labor in mice (Fig 1C), we only collected placental samples from women with term or spontaneous preterm labor for comparison of their ENaC expression. To avoid complications caused by Cesarean section, we only selected subjects that underwent vaginal delivery without Cesarean section.

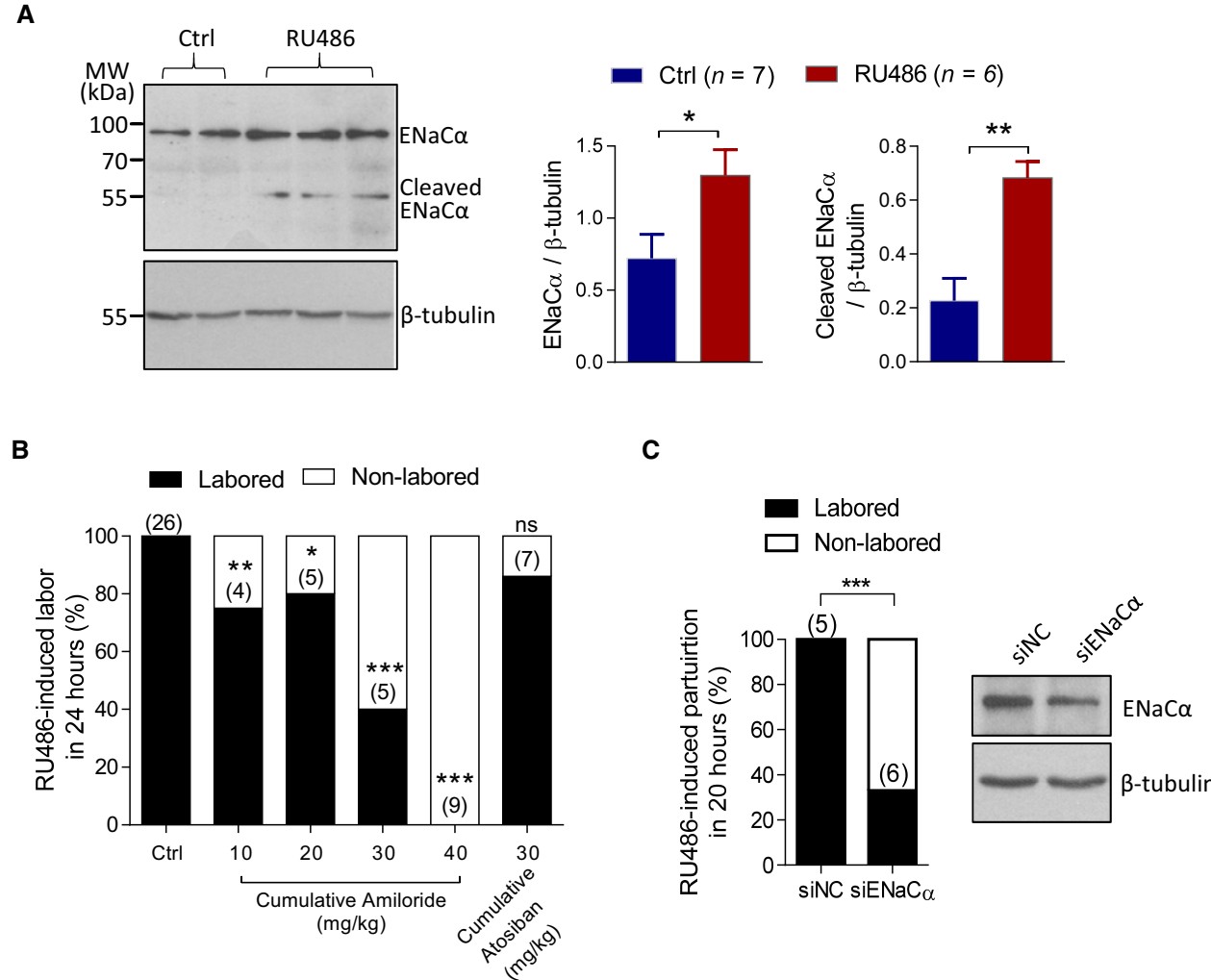

**Figure 4.  Inhibition of ENaC delays RU486-induced preterm labor in mice.**

A   Western blotting (left) with quantification (right) for ENaCα in uterine tissues collected from RU486 (200 μg per mouse, s.c.)-injected mice at the onset of preterm parturition and time-matched uterine tissues from vehicle-injected mice as control (Ctrl). Data are shown as mean ± SEM. *$P$ = 0.038, **$P$ = 0.0013, two-tailed unpaired Student's *t*-test, *n* is shown for each group.

B   Effect of cumulative injections of amiloride, atosiban (an antagonist of oxytocin receptor), or DMSO as vehicle control (Ctrl) on RU486-induced preterm labor in mice. Amiloride (initial dose, 10 mg/kg body weight) or atosiban (initial dose, 10 mg/kg body weight) was injected right before or every 7–8 h after RU486 injection for 1–4 times till the birth of the first pup. Data are percentages of mice labored within 24 h after RU486 injection. *n* is shown in each column. *$P$ < 0.05; **$P$ < 0.01; ***$P$ < 0.001, ns, not significant, chi-square test compared to Ctrl.

C   Effect of ENaCα knockdown on RU486-induced preterm labor in mice. siRNAs against ENaCα (siENaCα, 600 pmole) or non-silencing siRNAs as the control (siNC, 600 pmole) was injected (i.p.) in each mouse at 14 and 15 d.p.c. before RU486 injection. Data are percentages of mice labored within 20 h after RU486 injection. *n* is shown in each column. ***$P$ < 0.001, chi-square test. Right: Western blot for ENaCα in uterine tissues treated with siNC or siENaCα.

Data information: Exact *P*-values are listed in Appendix Table S1.
Source data are available online for this figure.

In addition, patients with genital infection were excluded. No obvious symptoms of intra-amniotic infection, such as unclear or bad smell amniotic fluid, were observed in the examined subjects. Clinical data are shown in Table 1. Western blot analysis showed significantly higher protein levels of ENaCα but not ENaCβ and γ in the placenta from women with preterm labor (*n* = 14) as compared to women labored at term (*n* = 15; Fig 5A and B). In addition, 86% of the subjects delivering preterm, and 13% delivering at term, had premature rupture of membranes (PROM). These PROM subjects showed higher ENaCα levels compared to those without PROM, although the difference was not statistically different (Fig EV4).

**ENaC is correlated with pro-inflammatory cytokines in women at labor**

Given the observed ENaC-dependent production of labor-associated pro-inflammatory factors in mouse uterus and ISK cells, we analyzed a previously published human transcriptome database of

**Table 1. Clinical characteristics.**

| Characteristics | Term (*n* = 15) | Preterm (*n* = 14) |
|---|---|---|
| Gravidity | 1.6 ± 0.7 | 1.5 ± 1.1 |
| Parity | 1.2 ± 0.4 | 1.2 ± 0.4 |
| Gestational weeks*** | 39.2 ± 0.7 | 33.6 ± 2.3 |
| Prenatal screening[a] | | |
| High risk | 1 (7%) | 1 (9%) |
| Low risk | 13 (93%) | 10 (91%) |
| Premature rupture of membrane | 2 (13%) | 12 (86%) |
| Vaginal delivery | 15 (100%) | 14 (100%) |
| Prenatal medication | | |
| Oxytocin use | 6 (40%) | 4 (28.6%) |
| Oxytocin dose (U) | 2.5 ± 0 | 4.4 ± 3.8 |
| Post-partum medication | | |
| Oxytocin use | 14 (93%) | 13 (93%) |
| Oxytocin dosage (U) | 11.4 ± 5.3 | 13.9 ± 5.1 |
| Number of newborns | | |
| Single birth | 15 (100%) | 13(93%) |
| Twins | 0 (0%) | 1 (7%) |
| Newborn gender | | |
| Female | 7 (47%) | 4 (27%) |
| Male | 8 (53%) | 11 (73%) |
| Birth weight (g)*** | 3,363.3 ± 401.5 | 2,268.6 ± 480.8 |
| Apgar score | | |
| First minute | 9.8 ± 0.8 | 9.3 ± 1.5 |
| Fifth minute | 10.0 ± 0 | 9.7 ± 0.8 |
| Neonatal complications | | |
| Hyperbilirubinemia | 1 (7%) | 7 (50%) |
| ABO hemolysis | 0 (0%) | 2 (14%) |
| Macrosomia | 1 (7%) | 0 (0%) |
| Asphyxia | 1 (7%) | 1 (7%) |
| Septicemia | 0 (0%) | 2 (14%) |

Values are means ± SD or numbers (percentages).
[a]Figures may not sum to total sample size because of missing data.
***$P < 0.001$, two-tailed unpaired Student's *t*-test.

maternal–fetus tissue/blood collected at onset of human term and preterm births (Data ref: Baldwin, 2015; preprint: Bukowski *et al*, 2017) for correlation between ENaC and pro-inflammatory factors. The results showed that in both preterm labor (*n* = 21, Fig 6A) and term labor (*n* = 38, Fig 6B) groups, ENaCα expression was found to be highly and positively correlated with pro-inflammatory factors (i.e., COX-2, IL-6, or IL-8), with stronger correlation observed in preterm labor compared to term labor (Fig 6A and B), suggesting an important role of ENaC in switching on inflammatory cytokines during labor in humans. More interestingly, the correlation between ENaC and pro-inflammatory factors found in the labor groups was not seen in non-labored birth groups, term or preterm (Fig 6C and D), further indicating the involvement of ENaC in labor process, term or preterm. Consistently, within in the same database, similar correlations were also seen in another labored birth group with preterm PROM (pPROM), but not in the non-labored pPROM group (Fig EV5).

## Discussion

The previously unsuspected role of ENaC in labor discovered in the present study provides an answer to how the mechanical signal is sensed and transduced into changes in cytokine profile, from anti-inflammatory to pro-inflammatory, and uterine contractility, from quiescent to a contractile state. The mechano-sensitivity of ENaC, as demonstrated in the present study by patch-clamp and Western blot showing cleaved form of ENaCα in uterine tissues upon stretching, and the capability of ENaC to transduce the mechanical signal into activation of the CREB/COX-2/PGE$_2$ axis render ENaC a suitable "sensor" and "switcher" required for labor. ENaC activation by embryo-released protease has been shown to result in membrane depolarization, Ca$^{2+}$ influx, and phosphorylation of CREB, a key transcription factor for COX-2, and thus enhanced production/release of PGE$_2$ required for embryo implantation (Ruan *et al*, 2012). Similar to its activation by protease, the present results show that the stretch-induced activation of ENaC also leads to phosphorylation of CREB and COX-2/PGE$_2$ (Fig 2). This explains the stretch-induced shift in cytokine profile during labor since the pro-inflammatory cytokines, such as IL-6, IL-8, and TNFα, are known to be the downstream targets of either CREB or PGE$_2$ (Wen *et al*, 2010; Srivastava *et al*, 2012). The involvement of ENaC in the regulation of pro-inflammatory factors during labor is further supported by the significant correlations between ENaC and COX-2, IL-6 or IL-8 found in the inflammatory transcriptome profiles at the maternal–fetal interface and onset of human preterm and term birth (Fig 6). Most intriguingly, such correlations are only found in the labor groups, term or preterm, but not in the non-labored birth groups (Fig 6). Taken together, the present results suggest that ENaC is an essential player for labor process, particularly for switching on a pro-inflammatory cytokine profile required for labor.

A positive-feedback loop for uterine contractions involving the production and release of uterine constrictive agonists, PGs, has long been proposed, mostly in myometrium muscle cells to explain the increasingly powerful expelling forces for labor (Wray, 1993; Lye *et al*, 2001). The presently demonstrated ENaC-dependent upregulation of COX-2 and increase in PGE$_2$ release upon stretch-enhanced uterine contractions provides a novel mechanism, with the new contribution of the decidua epithelial cells, underlying the putative positive-feedback loop of uterine contractions during parturition. Of note, we and others have also shown previously that PGE$_2$ exhibited similar role in mediating epithelial regulation of smooth muscles in other organ systems (Ruan *et al*, 2008, 2011). Interestingly, we have also shown previously that during embryo implantation, activation of ENaC can inhibit the COX-2 targeting microRNA, miR-199a-3p (Sun *et al*, 2014), which is reported to play important roles in labor as well (Renthal *et al*, 2013).

The involvement of ENaC in the process of labor is also supported by its upregulation toward the end of pregnancy and at labor in mice. This upregulation of ENaC could result from a couple of known initiating signals for parturition, including functional withdrawal of progesterone and increasing uterine stretch, since both

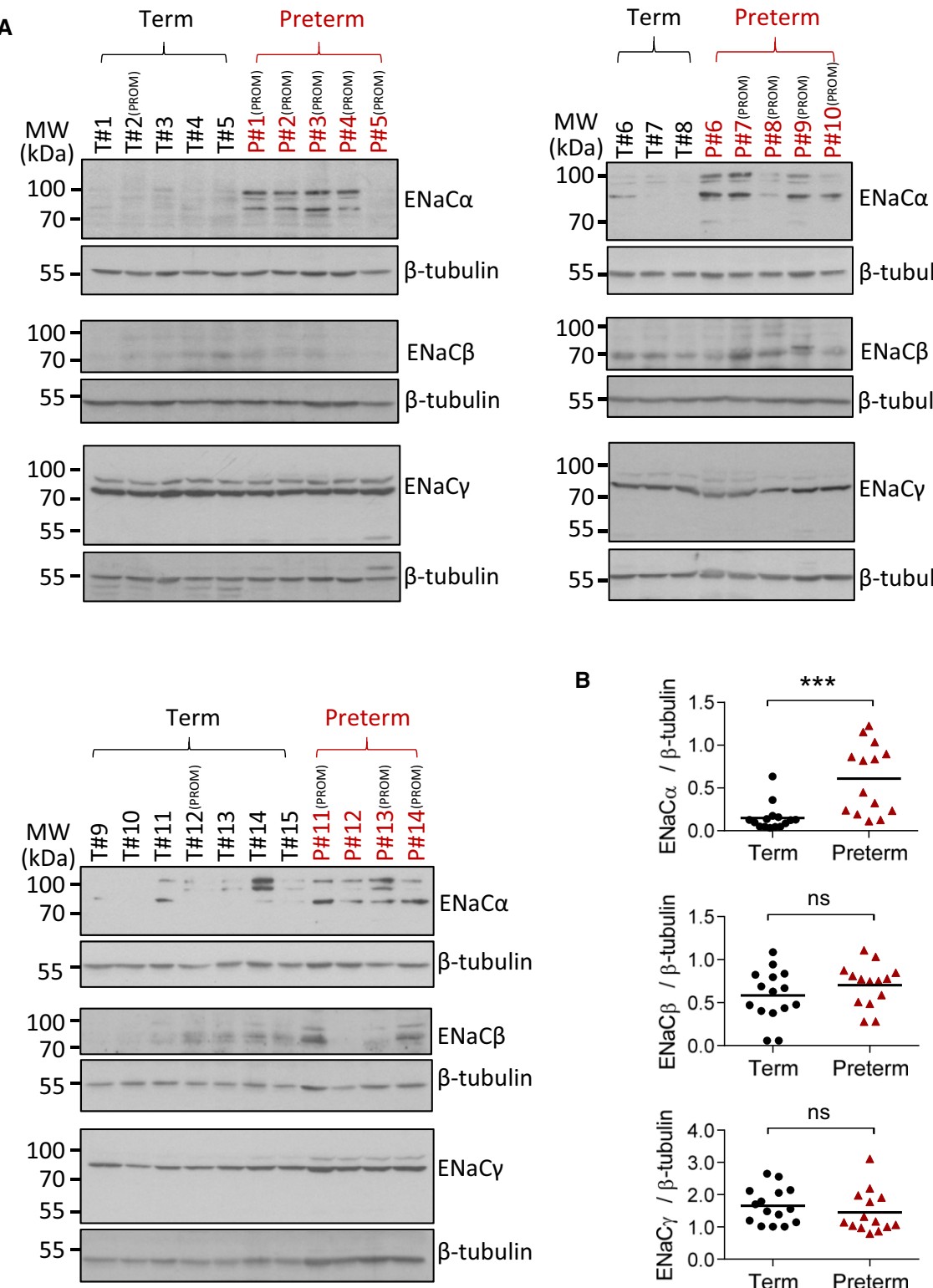

**Figure 5. ENaC expression is upregulated in women with spontaneous preterm labor.**

A, B    Western blotting for ENaCα, β, and γ (A) and their corresponding quantitation (B) in human placenta tissues collected at term (T#1–15) or spontaneous preterm (P#1–14) labor. ***$P < 0.001$, ns: $P > 0.05$, Mann–Whitney test. Clinical data are shown in Table 1. Subjects with premature rupture of membrane (PROM) are indicated. Exact $P$-values are listed in Appendix Table S1.

Source data are available online for this figure.

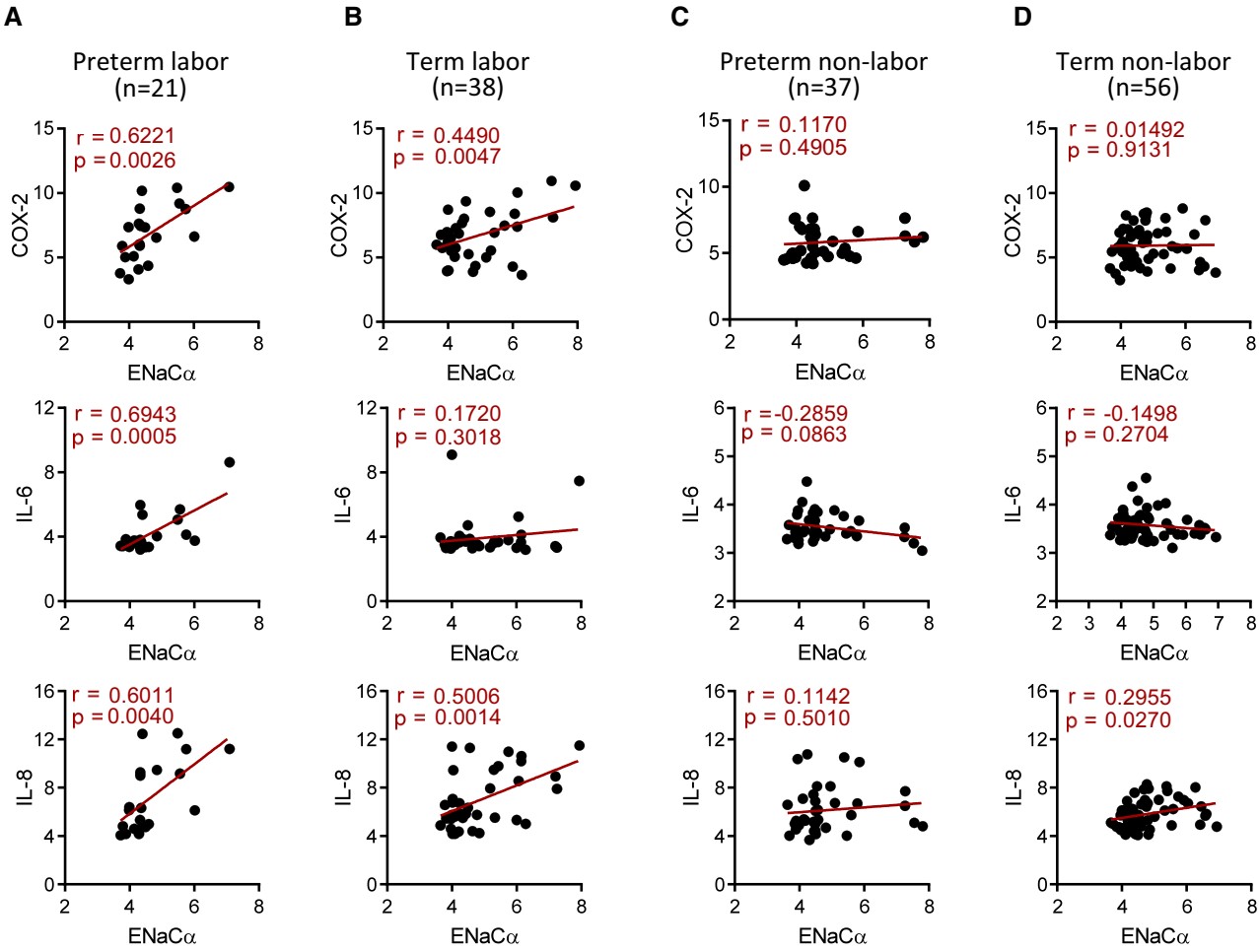

**Figure 6.  ENaC is correlated with pro-inflammatory cytokines in women at labor.**

A–D   Gene expression correlation analysis between ENaCα and COX-2, IL-6, and IL-8 in human maternal–fetus blood/tissue collected at onset of preterm (A and C) or term (B and D) birth labored (A and B) or non-labored (C and D). The microarray data were retrieved from a human database (Data ref: Baldwin, 2015). *n* is indicated for each group. Pearson correlation test and values of *r* and *P* are shown for each analysis.

were shown, in the present study, to promote ENaC expression in mouse uterine tissues. Disturbance of these signals is known to be associated with preterm labor. Progesterone functional withdrawal is considered a major cause for preterm labor, and its supplementation is currently used for most cases (Iams, 2014a). Multiple-fetus gestation is strongly associated with preterm labor, where uterine over-distension appears to be the mechanical trigger (Romero *et al*, 2014). In addition, maternal stress, which produces cortisol, a major glucocorticoid that enhances ENaC expression (Norlin *et al*, 1999), is also related to preterm labor (Petraglia *et al*, 2010). Of note, mammalian target of rapamycin (mTOR), known to activate ENaC (Almaca *et al*, 2013), has been reported to be increased in decidual senescence-caused preterm labor (Hirota *et al*, 2011; Cha *et al*, 2013), linking ENaC to another factor associated with preterm labor. ENaC appears to be associated with these recognized risk factors for preterm labor since ENaC can be untimely upregulated by these risk factors leading to pro-inflammatory shift in labor.

An important finding from the present study with translational implications is the use of ENaC blocker, amiloride, or siENaCα, to

delay RU486-induced preterm labor in mice. Interestingly, the effect of amiloride seems more potent than that of atosiban, a currently used drug to prevent preterm labor in humans (Worldwide Atosiban versus Beta-agonists Study Group, 2001; Saleh *et al*, 2013). Of note, amiloride does not seem to prevent preterm labor induced by LPS, a model mimicking infection-associated preterm labor (Fig EV3B). This is expected since LPS does not have any effect on ENaC upregulation (Fig EV3A). However, infection may directly activate COX-2-dependent inflammatory responses bypassing ENaC, suggesting that the LPS-induced preterm labor may be ENaC-independent. Interestingly, a recent study has demonstrated differential effects of an inhibitor of TRPV4 in delaying the delivery in the RU486 and LPS-induced preterm labor in mice, with predominant effect on the LPS-induced preterm labor model (Ying *et al*, 2015). This, together with the presently demonstrated differential effects of amiloride on the two preterm labor models, suggests different underlying mechanisms involved in these two preterm labor models.

Interestingly, human subjects with PROM in the present study seem to have a higher ENaCα level in average than non-PROM

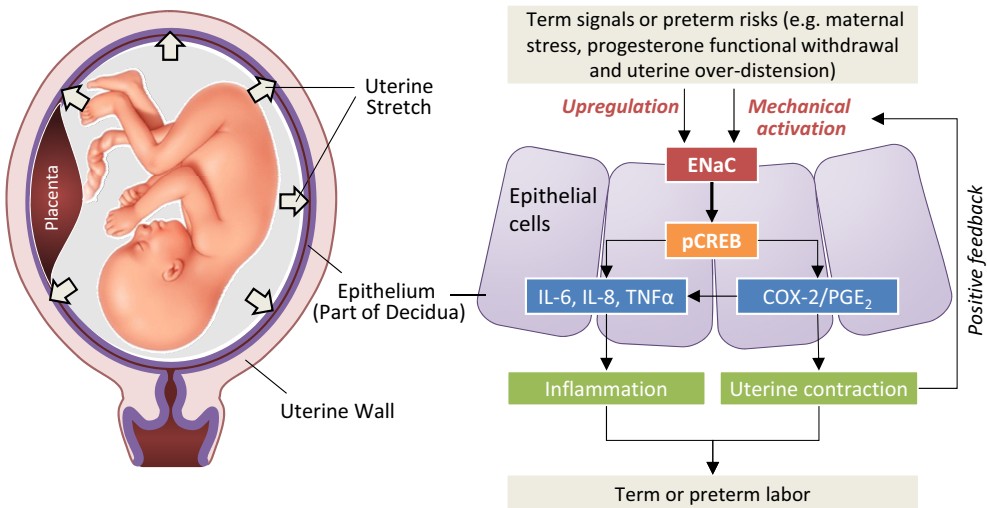

**Figure 7. The proposed role of ENaC in term and preterm labor.**

Term labor signals or preterm labor risk factors (e.g., maternal stress, progesterone functional withdrawal, uterine over-distension) may upregulate or mechanically activate ENaC in epithelial cells lining the uterine cavity (as part of the decidua), which results in phosphorylation of CREB (pCREB) and consequently upregulation of IL-6, IL-8, TNFα, and COX-2/PGE$_2$, leading to the switch from anti-inflammatory to inflammatory, quiescent to contractile state of the uterus for the initiation of labor.

subjects. While PROM is typically associated with infection, a number of studies have shown that infection is not always present in PROM (Romero *et al*, 1993; Shim *et al*, 2004; Menon & Fortunato, 2007; Goldenberg *et al*, 2008; Lannon *et al*, 2014; Menon & Richardson, 2017). Instead, inflammation, even in the absence of infection (e.g., chorioamnionitis), seems to be more consistently found in most PROM cases (Parry & Strauss, 1998; Shim *et al*, 2004; Romero *et al*, 2015; Waldorf *et al*, 2015). The sterile inflammation can be of multiple reasons, among which, over-distension of the uterus or stretching of the membrane triggering the production of pro-inflammatory factors is documented to be one key reason (Parry & Strauss, 1998; Waldorf *et al*, 2015). In the present study, women subjects with genital infections were excluded, and no obvious symptoms of intra-amniotic infection, such as unclear or bad smell amniotic fluid, were observed in examined subjects. Thus, while we cannot completely exclude the possibility of ascending infection in these subjects, our PROM cases did not have clinically discernible sign of infection. Given the presently demonstrated link of ENaC to uterine inflammation, especially stretch-induced inflammation, it is plausible that the higher ENaCα level in the PROM subjects is associated with sterile inflammation. In addition, since LPS was applied by intraperitoneal injection in our animal model, systematic overwhelming inflammation was induced, as indicated by extensive redness of intraperitoneal tissues, which may not exactly mimic all PROM in humans and would be very different from those only with local (e.g., intra-amniotic) and minor infection. The LPS endotoxin in the mouse model may bypass ENaC and induce overwhelming inflammation directly as reported (Wang & Hirsch, 2003).

Taken together, the present study has shed new light into the understanding of labor process. The demonstrated role of ENaC in term and preterm labor, as depicted in Fig 7, may have implications for treatment or prevention of preterm labor.

# Materials and Methods

### Mice

Adult female pregnant ICR (Institute of Cancer Research)/CD-1 mice were obtained from the Laboratory Animal Service Centre of the Chinese University of Hong Kong and maintained in a temperature-controlled room with a 12-h light-dark cycle, with food and water *ad libitum*. All animal experiments were carried out under guidelines on animal experimentation, and approval by the Animal Ethics Committee of the Chinese University of Hong Kong. The day a vaginal plug was found after mating was designated as day 1 post-coitum (d.p.c.). Animal sample size was estimated based on preliminary data and the observed effect sizes.

### Preterm labor in mice

RU486 (200 μg in 1 ml 5% ethanol) was injected subcutaneously (s.c.) into the flank region of each mouse at 15 d.p.c. to induce preterm labor. Amiloride (10 mg/kg body weight) was intraperitoneally injected into the mice right before and every 7–8 h after RU486 injection for up to four injections till the birth of the first pup. The cumulative doses of amiloride were therefore 10–40 mg/kg body weight. Atosiban (10 mg/kg body weight) was applied in the same fashion. Almost all of the atosiban-treated mice delivered before the 4th injection and thus the cumulative dose of atosiban was used up to 30 mg/kg body weight. In another set of experiments, lipopolysaccharide (LPS, 100 μg in 0.5 ml PBS) was intraperitoneally injected to each mouse at 15 d.p.c. to induce preterm labor as previously reported (Gross *et al*, 2000). The mice were sacrificed at the birth of one pup, and time-matched controls were killed right afterward. Animals were randomly grouped with different treatments, and animal studies were done without blinding.

### Mouse uterine tissue culture and stretch

Uterine tissues were collected from mice at 19 d.p.c. Two-cm (*in situ* length) uterine segments were longitudinally sliced into two halves. One half was stretched to *in situ* length (2 cm) and fixed with pins to a sylgard silicone-coated dish. The other half was pinned to the dish with slack length about 1 cm. These uterine preparations were cultured with DMEM supplemented with 10% FBS (v/v) and 1% penicillin–streptomycin (v/v) in 5% $CO_2$ incubators at 37°C for 1 h or overnight before tissues were collected for various analyses.

### Immunohistochemistry and immunofluorescence

Uterine tissues were fixed in 4% paraformaldehyde at 4°C overnight. For immunohistochemistry, tissues were embedded in paraffin and cut into 5-μm sections. Paraffin sections were deparaffinized, rehydrated, incubated with 3% hydrogen peroxide for 30 min to quench endogenous peroxidase activity and boiled in Tris–EDTA buffer (pH 9) for 10 min for antigen retrieval. Afterward, the sections were blocked in 2% normal horse serum for 30 min and incubated with antibody against ENaCα (1:100, Cat#ab77385, Abcam) at 4°C overnight. Subsequently, the Ultra Vision One HRP Polymer detection kit (Thermo Fisher Scientific) with DAB Plus Chromogen was used, and sections were counterstained with hematoxylin. For immunofluorescence, fixed uterine tissues were cryoprotected in 30% sucrose in PBS at 4°C for 24 h, embedded in OCT media (Tissue-Tek, 4583, Sakura), and cryo-sectioned into 5-μm sections. After microwaved in Tris–EDTA buffer (pH 9) for 20 min to retrieve antigens, treated with 1% SDS in PBS for 4 min and blocked with 1% bovine serum albumin in PBS for 15 min, sections were incubated with antibody against pCREB (1:200, Cat#9198, Cell Signaling Technology) overnight at 4°C and subsequently fluorochrome-conjugated secondary antibody (Invitrogen) for 1 h at room temperature. DAPI was used to stain cell nuclei. Quantification of imaging data was done using software ImageJ with blinding of the assessor.

### Cell culture

Human endometrial epithelial cell line Ishikawa (ISK) was purchased from ATCC (Virginia, United States) and cultured in RPMI-1640 supplemented with 10% fetal bovine serum (v/v) and 1% penicillin–streptomycin (v/v) in 5% $CO_2$ incubators at 37°C. The cell line was recently authenticated by STR profiling at the Department of Anatomical and Cellular Pathology, Faculty of Medicine, The Chinese University of Hong Kong.

### Gene knockdown

Lenti-virus (LV3) packaged shRNAs targeting human ENaCα (5′-GTG GCT GTG CCT ACA TCT TCT-3′) or scrambled non-coding shRNAs (5′-TTC TCC GAA CGT GTC ACG TTT-3′) were purchased from GenePharma (China). Viruses ($2 \times 10^7$ TU/ml) were transduced into ISK cells with polybrene (5 μg/ml). Cells were cultured in the presence of puromycin (5 μg/ml) for three passages to select stable clones before other experiments. siRNAs targeting to ENaCα (siENaCα, Cat#1320001, Assay-ID MSS208832) and non-silencing

siRNAs (siNC, Cat#12935300), Lipofectamine 2000 and Opti-MEM were purchased from Life Technologies. siENaCα (600 pmole) or siNC (600 pmole) combined with Lipofectamine 2000 in 500 μl Opti-MEM was intraperitoneally injected in each mouse at 14 and 15 d.p.c., respectively, before RU486 injection.

### Cell stretch

ISK cells were seeded in collagen type I-coated stretch chamber (ST-CH-04, B-Bridge International, Inc) and grown till 90% confluence. Cyclic (repetition of 30-s stretch and 30-s release) mechanical stretch by 15% elongation was applied for up to 1 h using the cell stretch device (Strex ST-150 Mechanical Cell Strain Instrument, B-Bridge International, Inc) in $CO_2$ incubator at 37°C before the cells were collected.

### Measurement of uterine contraction

ICR mice were sacrificed at 19 d.p.c. before the labor onset. After laparotomy, the uterus was exposed, and markings separated by 1 cm were made on the surface of the uterus before the uterus was excised, longitudinally cut open, removed of embryonic tissues, and cut into segments with *in situ* length of ~ 1 cm according to the markings. The *in situ* length of the preparations is defined as $L_0$, which is about 200% of the slack length of the excised uterine segments. Some segments were longitudinally cut into two halves with one half carefully peeled off the epithelium under a dissecting microscope and the other one kept intact. Comparisons were done between two halves of each pair. The uterine segmental preparations were mounted with one end attached to the bottom of a 37°C water-bath chamber and the other end to a silk thread, which connects to a force transducer. The preparation was subjected to a basal tension of 1 g and bathed a solution containing (in mM): 117 NaCl, 24.8 $NaHCO_3$, 4.7 KCl, 1.2 $MgCl_2$, 1.2 $KH_2PO_4$, 2.56 $CaCl_2$, 11.1 glucose, gassed with 95% $O_2$/5% $CO_2$, and equilibrated for 1 h during which time the bath solution was replaced with prewarmed fresh ones every 15 min. Isometric tension was recorded by a force transducer, and the output of the transducer was digitized by a signal collection and analysis system (BL-420E, Chengdu Technology & Market Co. Ltd, Chengdu, China). To apply mechanical stimuli, the uterine preparations were manually stretched using a micro-manipulator to achieve 10–40% elongation of $L_0$. The phase-advancing effect of stretches is calculated, according to a previously reported method (Kasai *et al*, 1995), as $(T_0-T_1)/(T_0-T_s)$, where $T_0$ is the time between onsets of two consecutive contractions prior to the stretch; $T_1$, period between two consecutive contractile onsets before and after the stretch; and $T_s$, the time when the stretch is initiated as measured from the onset of the previous contraction. The value of $(T_0-T_1)/(T_0-T_s)$ is 0 when $T_0 = T_1$ indicating no effect of stretch, and it increases and approaches a limiting value of 1 when $T_1 = T_s$ indicating immediate initiation of subsequent contraction upon stretch.

### RNA extraction and real-time PCR

Cells were lysed in TRIzol® Reagent (Invitrogen Life Technologies) according to manufacturer's instructions. Real-time PCR assays

were performed in triplicate on an Applied Biosystems 7500 Fast Real-Time PCR System. ENaCα (Hs00168906_m1, Mm00803391_m1) and COX-2 (Hs00153133_m1, Mm00478374_m1) analysis was performed with TaqMan probe (Invitrogen Life Technologies). SYBR green assay was used for human genes: IL-6 (Primers: *forward* 5′-GTCAGGGGTGGTTATTGCAT-3′ and *reverse* 5′-AGTGAGGAACAAGCCAGAGC-3′), IL-8 (Primers: *forward* 5′-TCT CTTGGCAGCCTTCCTG-3′ and *reverse* 5′-GAAGTTTCACTGGCATC TTCAC-3′), and TNFα (Primers: *forward* 5′-TCAGCCTCTTCTCC TTCCTG-3′ and *reverse* 5′-GCCAGAGGGCTGATTAGAGA-3′); and mouse genes: IL-6 (Primers: *forward* 5′-ACCAGAGGAAATTTTCAA TAGGC and *reverse* 5′-TGATGCACTTGCAGAAAACA-3′) and TNFα (Primers: *forward* 5′-ATGAGAGGGAGGCCATTTG-3′ and reverse 5′-CAGCCTCTTCTCATTCCTGC-3′). 18S rRNA or GAPDH was used as a housekeeping gene for normalization.

## Western blot

Cells or tissues were lysed in radioimmunoprecipitation assay (RIPA) lysis buffer (NaCl 150 mM, Tris–HCl (pH = 8.0) 50 mM, NP-40 1% v/v, sodium deoxycholate 0.5% w/v, and sodium dodecyl sulfate 0.1% w/v) with protease inhibitor cocktail (Cat# 88266, Thermo Scientific) and 1 mM phenylmethylsulphonyl fluoride. The lysates were analyzed by SDS–PAGE followed by probing antibodies. Antibodies against ENaCα (1:500, Cat#sc-22239, Santa Cruz Biotechnology, USA), ENaCβ (1:500, Cat#AB3532P, Millipore, USA), ENaCγ (1:1,000, Cat#ab3468, Abcam, England), and COX-2 (1:500, Catalog# 160106, Cayman Chemical, USA). For loading controls, antibodies against β-tubulin (Cat#sc-9104, Santa Cruz Biotechnology, USA) or GAPDH (Cat#sc-47724, Santa Cruz Biotechnology, USA) were used. The signal was detected with HRP-conjugated antibodies (Cat#170-6515, 170-6516, Bio-Rad, USA, and Cat#sc-2020, Santa Cruz Biotechnology, USA) and visualized by ECL detection reagents (Cat# RPN2106, RPN2232, GE Healthcare, UK). Densitometry of Western blots was performed by ImageJ software.

## $PGE_2$ ELISA

For $PGE_2$ detection in the medium, serum-free DMEM was used to incubate pretreated tissues for another 2 h before the medium was collected. EIA kit (Cat# 514010, Cayman Chemical) was used.

## Patch-clamp

An automatic patch-clamp system CytoPatch™ 2 (Cytocentrics AG, Rostock) was used according to the manufacturer's manual. Cells were suspended with a bath solution containing (in mM) 145 Na-gluconate, 2.7 KCl, 1.8 CaCl$_2$, 2 MgCl$_2$·6H$_2$O, 5.5 glucose, 10 HEPES, pH 7.4. Suspended cells in the bath solution were perfused into the double-channel CytoPatch™ microchip and captured by suction via the Cytocentering Channel in the chip. Afterward, suction via the CytoPatch™ Channel in the chip was applied to achieve > 10 GΩ seals to the cells before whole-cell mode was made. The cells were clamped at a voltage of −80 mV, and whole-cell currents were recorded by the electrode connected to the CytoPatch™ Channel in the chip. To apply stretching stimuli to the cells, negative pressures (40, 80, and 120 mPa) were applied through the Cytocentering

### The paper explained

#### Problem

Preterm labor, defined as birth before 37-week gestation in humans, represents a leading cause of neonatal death and disability. Despite its high incidence worldwide (5–18%), preterm labor remains hardly predictable and preventable largely due to the lack of understanding of the labor process. Given the similar clinical events involved, both term and preterm labor are considered to share common pathways of labor. The shift in cytokine profile from anti-inflammatory to pro-inflammatory and uterine activity from quiescent to contractile is the most recognizable sign of labor. However, the molecular mechanism underlying the "shift" of these events remains elusive.

#### Results

We report that the epithelial sodium channel (ENaC) is upregulated and activated in the uterus at labor in mice. Mechanical activation of ENaC results in phosphorylation of CREB and upregulation of pro-inflammatory cytokines as well as COX-2/PGE$_2$ in uterine epithelial cells. ENaC expression is also upregulated in mice with RU486-induced preterm labor as well as in women with preterm labor. Interference with ENaC attenuates mechanically stimulated uterine contractions and significantly delays the RU486-induced preterm labor in mice. Analysis of a human transcriptome database for maternal–fetus tissue/blood collected at onset of human term and preterm births reveals significant and positive correlation of ENaC with labor-associated pro-inflammatory factors in labored birth groups (both term and preterm), but not non-labored birth groups.

#### Impact

The study has shed new light into the understanding of labor process. The demonstrated role of ENaC in transducing mechanical signal into cytokine profile shift to pro-inflammatory in term and preterm labor may suggest it as a potential molecular target for treatment or prevention of preterm labor.

Channel in the chip. The intracellular solution contained (in mM): 135 K-gluconate, 10 KCl, 6 NaCl, 2 MgCl$_2$.6H$_2$O, 10 HEPES, pH 7.2.

## Human placental tissue collection

Human placenta samples were obtained from women at Women's Hospital of School of Medicine of Zhejiang University. The study was approved by the Ethics Committee of Women's Hospital of School of Medicine of Zhejiang University, and patients gave their written informed consent. Women with genital infection were excluded. All the subjects underwent vaginal delivery without Cesarean section. The experiments conformed to the principles set out in the WMA Declaration of Helsinki and the Department of Health and Human Services Belmont Report.

## Statistical analysis

The software GraphPad Prism 6.0 was used for graphing and statistical analyzing the data. Data are shown as mean ± SEM. Two-tailed Student's paired or unpaired *t*-test was used for two-group comparison. One-way ANOVA was used for comparing three or more groups. Chi-square test was used for categorical variables. Pearson test was used for correlation analysis. Mann–Whitney test as was used a nonparametric test. $P < 0.05$ was considered as statistically significant.

The variance was calculated to be similar between comparison groups or otherwise corrected for particular tests.

## Data availability

Previously published human transcriptome datasets from maternal–fetus interface tissues/blood collected at onset of preterm and term birth (Data ref: Baldwin, 2015; Bukowski *et al*, 2017) were analyzed for correlations between ENaC and inflammatory factors.

**Expanded View** for this article is available online.

## Acknowledgements

The work was supported in part by General Research Fund of Hong Kong (Y.C.R. No. 14112814), National 973 Projects (H.C.C. 2013CB967401, 2013CB967404), Natural Science Foundation of China (Y.C.R, No. 81471460; J.H.G, No. 81300515; H.C.C. No. 81370709), Early Career Scheme (Y.C.R. No. 24104517) and Theme-based Research Scheme of Hong Kong (Y.C.R. No. T13-402/17N), and Start-up fund at the Hong Kong Polytechnic University. We thank Linling Zhu, Peng Xu, Yu-Jia Kong, and Meng-Yan Xu for their help with collecting clinical samples. We thank Yi-Mi Xie for help with image quantification. We thank Dr Kun-Han Lin for his help with CytoPatch experiments.

## Author contributions

HCC, YCR, and JHG: conception and design. XS, JHG, J-jC, WYL, HC, WQH, LLT, MKY, XJ, YWC, and YCR: experiments and/or data analysis. DZ, YH, YL, and HH: clinical materials and consultancy. XS, YCR, and HCC: article writing with contribution from other authors.

## Conflict of interest

The authors declare that they have no conflict of interest.

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
