## [Review Process File · EMBO Molecular Medicine]

Activation of the epithelial sodium channel (ENaC) leads to cytokine profile shift to pro-inflammatory in labor

Xiao Sun, Jing Hui Guo, Dan Zhang, Jun-jiang Chen, Wei Yin Lin, Yun Huang, Hui Chen, Wen Qing Huang, Yifeng Liu, Lai Ling Tsang, Mei Kuen Yu, Yiu Wa Chung, Xiaohua Jiang, Hefeng Huang, Hsiao Chang Chan and Ye Chun Ruan

Review timeline:

Submission date:	12 January 2018
Editorial Decision:	21 February 2018
Revision received:	23 May 2018
Editorial Decision:	03 July 2018
Revision received:	16 July 2018
Accepted:	24 July 2018

Editor: Céline Carret

Transaction Report:

1st Editorial Decision

21 February 2018

Thank you for the submission of your manuscript to EMBO Molecular Medicine and apologies for the usual delay in getting back to you, due to staff travelling and initial difficulties in securing three referees. We have now heard back from the referees whom we asked to evaluate your manuscript.

As you will see from the set of comments pasted below, all referees find the paper to be of interest. However, better statistics, animal description, providing n= and dosage of drugs is mandatory and should be provided in all instances (also as per our guidelines), along with better blots and a language/grammar review. You will see that ref. 1 and 3 highlight important potential conceptual flaws in 1) role of inflammation vs. progesterone withdrawal in PROM and consequences of preterm births and 2) that no causative role of ENaC in preterm birth is demonstrated. Upon our cross-commenting exercise, referees 1 and 2 both agreed that showing a causative role for ENaC up-regulation in pre-term birth in vivo would be beyond the scope of the manuscript. As such, we would like to encourage you to address all the remaining issues, including number 1) above but not focus on 2). Please follow referee 3's recommendations in rephrasing title, abstract, and introduction accordingly to tune down claims of any causative effect.

We would welcome the submission of a revised version within three months for further consideration and would like to encourage you to address all the criticisms raised as suggested to improve conclusiveness and clarity. Please note that EMBO Molecular Medicine strongly supports a single round of revision and that, as acceptance or rejection of the manuscript will depend on another round of review, your responses should be as complete as possible.

I look forward to receiving your revised manuscript.

***** Reviewer's comments *****

Referee #1 (Remarks for Author):

This manuscript addresses a very important clinical issue, as preterm birth accounts for a tremendous amount of perinatal morbidity and mortality. As most investigators in the field are focused on other aspects of preterm birth, this paper looks at the problem from a refreshing angle. However, I have a few concerns about this manuscript.

The first issue below is the most serious concern.

The authors conclude from data shown in Figure S2 that ENaC is involved in preterm labor caused by progesterone withdrawal, but not in preterm labor related to the pro-inflammatory response induced by LPS. The data in Figure 5 suggest that ENaC is involved in preterm labor in human subjects. Most of the human subjects delivering preterm (86%) in the study had premature rupture of membranes (PROM), while only 13% of the subjects delivering at term had PROM. But inflammation secondary to bacterial infection, similar to that which LPS would induce, is often the cause of PROM, not progesterone withdrawal and not uterine stretching. How do you reconcile this paradox? Eliminating patients with genital infection would not eliminate patients with ascending bacterial infections and inflammation typically associated with PROM.

The quality of the Western blots is poor and there is a great deal of variability among the biological replicates throughout the paper. In particular, the quantification of the results shown in Figure 2h should be included.

There are many language errors in the paper, too many for me to list. Some important ones are:

"and so as in women" should be "as well as in women" in the abstract;

"could be resulted from" should be "could result from" in line 292;

"withdrawn of" should be "withdrawal of" in line 293;

I think "peered" should be "peeled" in line 403.

I recommend that the paper be reviewed and corrected separately for grammatical errors.

Referee #2 (Remarks for Author):

The manuscript by Sun and colleagues investigates a novel role of the epithelial sodium channel ENaC for preterm labor in a mouse model. The authors find a gradual increase in ENaC expression in murine uterine tissues during late gestation which is accompanied by elevated ENaC activation. Due to the previously established mechano-sensitivity of ENaC, the authors examined a potential contribution of ENaC to mechanical force (stretch)-induced activation of downstream transcription factors (CREB), proinflammatory cytokines and prostaglandins using pharmacological inhibition of ENaC with amiloride in a human endometrial epithelial cell line. ENaC inhibition through amiloride also delayed stretch-induced uterine contractions in vivo. In addition, the authors show that RU486- or LPS-induced preterm labor can be effectively prevented by both siRNA-mediated ENaC knockdown in vivo as well as pharmacological inhibition. At last, the authors also provide human data by means of a gene expression correlation analysis that supports their conclusions.

This is a very interesting study with robust data and solid experimental work. The manuscript is well-written and features a variety of established methods that are all well controlled. Except for a few instances stated below proper statistics were adequately applied. The following points need to be addressed in a revised version.

Major points:

1. Figure 1A: Statistical analysis (t-test results) should be indicated in the figure panel by using asterisks.
2. Figure 2H: CREB/Phosphorylated CREB should both be included in the immunoblot analysis.
3. Supplementary Figures S2A+B: The number of animals used in this experiment should be included in the figure legend.
4. Supplementary Figure S2B: Statistical analysis should be applied and indicated within figure and/or legend.

Referee #3 (Remarks for Author):

The paper submitted to EMBO Journal focuses on the role of the epithelial sodium channel (ENaC) in the process of parturition, with a special attention to its relation to preterm labor. The authors report that its mechano-activation leads to a shift in the cytokine profile from anti- to pro-inflammatory through the CREB/COX-2/PGE2 axis that can induce delivery and perpetuate labor initializing a positive feedback loop, and throughout the paper suggest that abnormalities of the channel, either in expression or function, result in preterm labor.

The paper is well-written, well-organized and follows the EMBO Journal guidelines. The authors performed many different experiments with mice, human cell lines and human tissue samples. The methodology is accurately described, and the results are presented clearly and objectively. There is a solid amount of data shown that supports the author's conclusions about the role of the ENaC in labor.

However, there is no data at all to sustain the hypothesis that abnormalities of the channel lead to preterm labor, as suggested throughout the paper. The upregulation and its consequences, like the activation of the CREB/COX-2/PGE2 axis, seen in both preterm induced mice and human samples from preterm patients, don't seem to have a causative effect, but to simply be a part of the process of labor itself. As the authors themselves state in the introduction, both term and preterm labor are considered to share common pathways, like the shift in cytokine profile from anti- to pro-inflammatory and the change in the uterine activity from quiescent to contractile. Thus, it is no surprise that the same increase in transcription and activity of the ENaC that can be seen in term labor can also be detected in preterm labor.

Furthermore, to test the hypothesis that the channel is abnormally upregulated in preterm labor it would be necessary to compare the expression and function of the channel in preterm labor to those in term labor, and not to those in time-matched controls that are not in labor at all, like the authors did. Only if significant differences between the two groups existed, it could be defined as abnormal upregulation. Also, to be able to define it as a cause for preterm labor, it would have to be proven that an abnormal upregulation of the channel can induce preterm labor. Otherwise, it appears to just be the normal upregulation that happens to take place during labor, but just earlier than in term deliveries, with the cause of the earliness remaining unknown. The higher expression of ENaC α found in placentas from women with spontaneous preterm labor as compared to women with term labor is not enough data to sustain the hypothesis.

Therefore, the upregulation and activity of the channel in preterm labor must not be defined as a cause for it, but rather as part of the process, like in term labor. It still is, though, an interesting and valuable conclusion, as it unravels a previously unknown mechanism that acts in both term and preterm labor, and additionally could potentially serve as a target for the prevention and treatment of preterm labor.-

Suggested revisions

Title: The title is misleading since it gives the impression that an abnormal upregulation of the ENaC can be the cause of preterm labor, hypothesis for which there is no data to support it, as discussed above. Therefore, it should be changed. As proposed title it could be: Activation of the epithelial sodium channel (ENaC) in labor leads to a shift in the cytokine profile from anti- to pro-inflammatory.

Abstract: The abstract should be refocused around the general role of the ENaC in labor, and the parts about the suspected causality in preterm labor should be taken out. Also, it's not necessary to center that much on preterm labor, since the vast majority of the paper is about term labor, with preterm labor only accounting for a small part of it.

Introduction: As in the abstract, the focus shouldn't be that much on preterm labor, but rather the process of labor itself (mostly first paragraph).

Results: Interference of ENaC delays RU486-induced preterm labor in mice: Dosages of amiloride and atosiban should be clarified, as they are not easily understandable and the description of the methodology seems incomplete (see methods), and it may be interesting to compare with more than just one dose of atosiban. Also, data must be statistically analyzed; especially when it is later discussed that amiloride is more potent than atosiban. Line 191: The upregulation of ENaC by RU486 is not the cause for the preterm labor, but rather just indicates an involvement of the ENaC in progesterone withdrawal-related preterm labor.

Discussion: Line 311: It can't be stated that the effect of amiloride is more potent than that of atosiban if the data is not statistically analyzed (see results). Line 326: There is no data to draw any conclusions about the cause of spontaneous preterm labor (as discussed).

Methods: Mice: Exact species and origin of the mice should be clarified. Preterm labor in mice: Dosages of amiloride should be clarified (which were the different initial and cumulative doses used for comparison), and the administration of atosiban and its exact dosage should be mentioned as well (as for amiloride). Also, on lines 344-5 it says that some 15 d.p.c mice were daily injected with progesterone for 2 days, but neither the reason nor the results are stated anywhere. That should be added. Statistical analysis: The program used for analysis and graphing of the data should be mentioned.

Figures and Tables: Figure 1a: Statistical analysis should be performed. Figure 4b: Statistical analysis has to be done. Also, there is only one dosage of amiloride mentioned in the description while four different were used, and it is not clear if the ones in the graph are initial or cumulative. For atosiban it should be clarified if the dosage in the graph is cumulative as well. Table 1: Asterisk (* or even ***) should be used for $p < 0.001$ for easier understanding.

Minor revisions

Line 61-2: It should say, "A large number of studies are dedicated to myometrium muscle cells and suggest..."

Line 66: "Are" should be changed to "is" ("...is not well studied"). Also, there are two spaces before the "are".

Line 98: There is a period in the title which is not necessary.

Line 342-3: It is not necessary to state that none of the vehicle-injected mice delivered prematurely in Methods, since this is data that is part of the results.

Line 480-1: It is not necessary to refer to Table 1 in Methods, since the data shown there is part of the results.

1st Revision - authors' response

23 May 2018

Referee #1 (Remarks for Author):

This manuscript addresses a very important clinical issue, as preterm birth accounts for a tremendous amount of perinatal morbidity and mortality. As most investigators in the field are focused on other aspects of preterm birth, this paper looks at the problem from a refreshing angle. However, I have a few concerns about this manuscript.

The first issue below is the most serious concern.

The authors conclude from data shown in Figure S2 that ENaC is involved in preterm labor caused by progesterone withdrawal, but not in preterm labor related to the pro-inflammatory response induced by LPS. The data in Figure 5 suggest that ENaC is involved in preterm labor in human subjects. Most of the human subjects delivering preterm (86%) in the study had premature rupture of membranes (PROM), while only 13% of the subjects delivering at term had PROM. But inflammation secondary to bacterial infection, similar to that which LPS would induce, is often the cause of PROM, not progesterone withdrawal and not uterine stretching. How do you reconcile this paradox? Eliminating patients with genital infection would not eliminate patients with ascending bacterial infections and inflammation typically associated with PROM.

Answer:

We thank the reviewer for pointing out this very important issue. We did further analysis and confirmed that the PROM subjects did have higher ENaC α expression levels as compared to those with non-PROM, although the difference is not statically significant (Reply Fig.R1, Supplementary Fig.S4 in the revision).

We agree with the reviewer that infection is typically associated with PROM, however, it should also be noted that a number of literatures have indicated that infection is not always present in all or majority of PROM cases 1-6. Some even debated whether infection is a cause or consequence of PROM⁵, since PROM has been documented in many cases even after effective antibacterial therapy. Instead, inflammation, even in the absence of infection (e.g. chorioamnionitis), seems to be more consistently found in most PROM cases⁶⁻⁹. The sterile inflammation can be of multiple reasons, among which, over-distension of the uterus or stretching of the membrane triggering the production of proinflammatory factors is documented to be one key reason^{8,9}.

In our study, as most of the preterm cases have PROM, we double checked with the clinicians who collected the samples. Our clinicians have confirmed that no obvious symptoms of intra-amniotic infection, such as unclear or bad smell amniotic fluid, were observed in these subjects. Otherwise, definitive microbial tests of the amniotic fluid would have been done. Thus, while we cannot completely exclude the possibility of ascending infection in these subjects as the reviewer pointed out, our PROM cases did not have clinically discernible sign of infection. On the other hand, it is interesting to note that the present study has in fact provided evidence to link ENaC to uterine inflammation, especially stretch-induced inflammation. Thus, it is plausible that the higher ENaC α level in PROM subjects is associated with sterile inflammation.

In addition, since LPS was applied by intraperitoneal injection in our animal model, systematic overwhelming inflammation was induced, as indicated by extensive redness of intraperitoneal tissues and death of foetuses, which is very different from the RU486 model where no severe tissue redness was seen and foetuses were mostly alive when delivered. The LPS model, therefore, may not exactly mimic all PROM in humans and would be very different from those only with local (e.g. intra-amniotic) and minor infection. The LPS endotoxin in the mouse model may bypass ENaC and induce overwhelming inflammation directly as reported¹⁰.

Thanks again for bring up this important issue and we included all above discussed points in the discussion of the revised manuscript (lines 345-368, page 12). The additional analysis and detailed description of the subjects and models have also been included (lines 227, 244, 250, page 8-9).

Reply Fig.R1. Quantification of western blotting for ENaC α in placental tissues from women labored at term or preterm with or without premature rupture of membrane (PROM). ns: $p > 0.05$, Mann-Whitney test.

The quality of the Western blots is poor and there is a great deal of variability among the biological replicates throughout the paper. In particular, the quantification of the results shown in Figure 2h should be included.

Answer:

Thanks for the comment and suggestion. For all of the western blots, we have reviewed our original data. Poor quality images have been replaced with clearer or more representative ones. Some of the experiments were repeated to gain more conclusive data. Western blot data in Figure 2H have been quantified and shown on the right panel in Figure 2H in the revision.

There are many language errors in the paper, too many for me to list. Some important ones are:

"and so as in women" should be "as well as in women" in the abstract;

"could be resulted from" should be "could result from" in line 292;

"withdrawn of" should be "withdrawal of" in line 293;

I think "peered" should be "peeled" i line 403.

I recommend that the paper be reviewed and corrected separately for grammatical errors.

Answer:

We thank the reviewer for spotting these language errors. We have corrected them accordingly. The manuscript has also been reviewed for other grammatical errors.

Referee #2 (Remarks for Author):

The manuscript by Sun and colleagues investigates a novel role of the epithelial sodium channel ENaC for preterm labor in a mouse model. The authors find a gradual increase in ENaC expression in murine uterine tissues during late gestation which is accompanied by elevated ENaC activation. Due to the previously established mechano-sensitivity of ENaC, the authors examined a potential

contribution of ENaC to mechanical force (stretch)-induced activation of downstream transcription factors (CREB), proinflammatory cytokines and prostaglandins using pharmacological inhibition of ENaC with amiloride in a human endometrial epithelial cell line. ENaC inhibition through amiloride also delayed stretch-induced uterine contractions in vivo. In addition, the authors show that RU486- or LPS-induced preterm labor can be effectively prevented by both siRNA-mediated ENaC knockdown in vivo as well as pharmacological inhibition. At last, the authors also provide human data by means of a gene expression correlation analysis that supports their conclusions.

This is a very interesting study with robust data and solid experimental work. The manuscript is well-written and features a variety of established methods that are all well controlled. Except for a few instances stated below proper statistics were adequately applied. The following points need to be addressed in a revised version.

Major points:

1. Figure 1A: Statistical analysis (t-test results) should be indicated in the figure panel by using asterisks.

Answer: Thanks for the suggestion. Statistical analysis has been performed for figure 1A and results are shown with asterisks in the revision.

2. Figure 2H: CREB/Phosphorylated CREB should both be included in the immunoblot analysis.

Answer: Thanks for the suggestion. We did examine phosphorylated CREB (pCREB) in the stretched uterine tissues by immunoblot, which however did not show significant increase after stretch. This is probably due to the uterine tissues are of heterogeneous cell populations. The immunofluorescence labelling of tissues after stretch showed pCREB signals lighten up mainly in epithelial cells (Figure 2A), but not other cells (e.g. smooth muscles). Homogenized tissues for immunoblot may mask the phosphorylated CREB in epithelial cells. For this reason, we did additional experiment to perform the stretch in the endometrial epithelial cell line. The result showed that amiloride repressed stretch-induced increase of pCREB, validating ENaC-mediated elevation of pCREB after stretch in these cells (Reply Fig.R2, Supplementary Fig.S1).

Reply Fig. R2. Stretch induces ENaC-dependent CREB activation in human endometrial epithelial cells. Western blotting with quantification for pCREB in ISK cells with (+) or without (-) 30 min stretch, in the absence (-) or presence (+) of Ami (10 μ M). * p <0.05, One-way ANOVA with Tukey's multiple comparisons test.

3. Supplementary Figures S2A+B: The number of animals used in this experiment should be included in the figure legend.

Answer: Thanks for the suggestion. The number of animals have been included.

4. Supplementary Figure S2B: Statistical analysis should be applied and indicated within figure and/or legend.

Answer: Thanks for the suggestion. Statistical analysis has been performed.

Referee #3 (Remarks for Author):

The paper submitted to EMBO Journal focuses on the role of the epithelial sodium channel (ENaC) in the process of parturition, with a special attention to its relation to preterm labor. The authors report that its mechano-activation leads to a shift in the cytokine profile from anti- to pro-inflammatory through the CREB/COX-2/PGE2 axis that can induce delivery and perpetuate labor initializing a positive feedback loop, and throughout the paper suggest that abnormalities of the channel, either in expression or function, result in preterm labor.

The paper is well-written, well-organized and follows the EMBO Journal guidelines. The authors performed many different experiments with mice, human cell lines and human tissue samples. The methodology is accurately described, and the results are presented clearly and objectively. There is a solid amount of data shown that supports the author's conclusions about the role of the ENaC in labor.

However, there is no data at all to sustain the hypothesis that abnormalities of the channel lead to preterm labor, as suggested throughout the paper. The upregulation and its consequences, like the activation of the CREB/COX-2/PGE2 axis, seen in both preterm induced mice and human samples from preterm patients, don't seem to have a causative effect, but to simply be a part of the process of labor itself. As the authors themselves state in the introduction, both term and preterm labor are considered to share common pathways, like the shift in cytokine profile from anti- to pro-inflammatory and the change in the uterine activity from quiescent to contractile. Thus, it is no surprise that the same increase in transcription and activity of the ENaC that can be seen in term labor can also be detected in preterm labor.

Furthermore, to test the hypothesis that the channel is abnormally upregulated in preterm labor it would be necessary to compare the expression and function of the channel in preterm labor to those in term labor, and not to those in time-matched controls that are not in labor at all, like the authors did. Only if significant differences between the two groups existed, it could be defined as abnormal upregulation. Also, to be able to define it as a cause for preterm labor, it would have to be proven that an abnormal upregulation of the channel can induce preterm labor. Otherwise, it appears to just be the normal upregulation that happens to take place during labor, but just earlier than in term deliveries, with the cause of the earliness remaining unknown. The higher expression of ENaC α found in placentas from women with spontaneous preterm labor as compared to women with term labor is not enough data to sustain the hypothesis.

Therefore, the upregulation and activity of the channel in preterm labor must not be defined as a cause for it, but rather as part of the process, like in term labor. It still is, though, an interesting and valuable conclusion, as it unravels a previously unknown mechanism that acts in both term and preterm labor, and additionally could potentially serve as a target for the prevention and treatment of preterm labor.

Answer:

We thank the reviewer very much for careful analysis of our manuscript and recognition of the key message we intend to deliver: ENaC is a previously unknown mechanism both in term and preterm labor, and could be a potential target for the prevention and treatment of preterm labor.

The reviewer's concern on causative effect of ENaC is well taken. We have changed the title and rephrased the statements throughout the manuscript to avoid the claim of causative role of ENaC according to the reviewer's suggestions.

Suggested revisions

Title: The title is misleading since it gives the impression that an abnormal upregulation of the ENaC can be the cause of preterm labor, hypothesis for which there is no data to support it, as

discussed above. Therefore, it should be changed. As proposed title it could be: Activation of the epithelial sodium channel (ENaC) in labor leads to a shift in the cytokine profile from anti- to pro-inflammatory.

Answer: Thanks for the suggestion. This important point is well taken. We have modified the title into: Activation of the epithelial sodium channel (ENaC) leads to cytokine profile shift to pro-inflammatory in labor.

Abstract: The abstract should be refocused around the general role of the ENaC in labor, and the parts about the suspected causality in preterm labor should be taken out. Also, it's not necessary to center that much on preterm labor, since the vast majority of the paper is about term labor, with preterm labor only accounting for a small part of it.

Answer: Thanks. We agree to refocus the general role of ENaC in labor. We have modified abstract and introduction accordingly.

Introduction: As in the abstract, the focus shouldn't be that much on preterm labor, but rather the process of labor itself (mostly first paragraph).

Results: Interference of ENaC delays RU486-induced preterm labor in mice: Dosages of amiloride and atosiban should be clarified, as they are not easily understandable and the description of the methodology seems incomplete (see methods), and it may be interesting to compare with more than just one dose of atosiban. Also, data must be statistically analyzed; especially when it is later discussed that amiloride is more potent than atosiban. Line 191: The upregulation of ENaC by RU486 is not the cause for the preterm labor, but rather just indicates an involvement of the ENaC in progesterone withdrawal-related preterm labor.

Answer: Thanks. We have modified the methods, results and legends to make the description of dosage clear (line 205-206, page 7; line 384-390, page 14; line 856-859, page 27). Relevant data have been statistically analysed and results show significant difference between control and amiloride-treated, but not between control and atosiban-treated mice, which has been included in the revision.

The dosage of atosiban suggested for clinical use in humans is up to 270 mg per person, or 4-5 mg/kg body weight (<https://www.medicines.org.uk/emc/product/7056/smpc>). What we used in Figure 4, cumulatively 30 mg/kg, is a much higher dose. We did try a lower dose of 4.12 mg/kg as previously reported¹¹, which however did not show significant effect.

Discussion: Line 311: It can't be stated that the effect of amiloride is more potent than that of atosiban if the data is not statistically analyzed (see results).

Answer: Thanks. This point is well taken. Related statements have been modified (line 196, page 7).

Line 326: There is no data to draw any conclusions about the cause of spontaneous preterm labor (as discussed).

Answer: The data have been statistically analyzed. Amiloride (10-40 mg/kg)-treated, but not atosiban (30 mg/kg)-treated showed significant difference with the control mice (Chi square test).

Methods: Mice: Exact species and origin of the mice should be clarified. Preterm labor in mice: Dosages of amiloride should be clarified (which were the different initial and cumulative doses used for comparison), and the administration of atosiban and its exact dosage should be mentioned as well (as for amiloride).

Answer: The mouse strain we used is ICR (Institute of Cancer Research)/CD1, which has been clarified (line 374, page 14). Detailed description of the dosage of amiloride/atosiban are now included in the Methods (line 389, page 14).

Also, on lines 344-5 it says that some 15 d.p.c mice were daily injected with progesterone for 2 days, but neither the reason nor the results are stated anywhere. That should be added.

Answer: Thanks for spotting this mistake. We have removed this sentence as relative experiments are not included in the present study.

Statistical analysis: The program used for analysis and graphing of the data should be mentioned.

Answer: GraphPad Prism 6.0 was used to analyze and graph all the data. This has been included in Methods (line 544, page19).

Figures and Tables: Figure 1a: Statistical analysis should be performed.

Answer: Thanks for the suggestion. Statistical analysis has been performed for figure 1A and results are shown with asterisks in the revision.

Figure 4b: Statistical analysis has to be done. Also, there is only one dosage of amiloride mentioned in the description while four different were used, and it is not clear if the ones in the graph are initial or cumulative. For atosiban it should be clarified if the dosage in the graph is cumulative as well.

Answer: Thanks for the suggestion. The statistical analysis has been done. The dosage of amiloride and atosiban has been clarified in results, methods, figures and legends (line 205-206, page 7; line 384-390, page14; line856-859, page 27).

Table 1: Asterisk (* or even ***) should be used for $p < 0.001$ for easier understanding.

Answer: Thanks. We have modified the table accordingly. *** is used for $p < 0.001$.

Minor revisions

Answer: We thank the reviewer again for careful review of our manuscript and pointing out these errors. All the minor points have been revised accordingly.

Line 61-2: It should say, "A large number of studies are dedicated to myometrium muscle cells and suggest..."

Answer: Revised (line 65). Thanks.

Line 66: "Are" should be changed to "is" ("...is not well studied"). Also, there are two spaces before the "are".

Answer: Revised (line 65). Thanks.

Line 98: There is a period in the title which is not necessary.

Answer: Removed the period (line 103). Thanks.

Line 342-3: It is not necessary to state that none of the vehicle-injected mice delivered prematurely in Methods, since this is data that is part of the results.

Answer: Removed. Thanks.

Line 480-1: It is not necessary to refer to Table 1 in Methods, since the data shown there is part of the results.

Answer: Removed. Thanks.

References

1. Romero, R. *et al.* The relationship between spontaneous rupture of membranes, labor, and microbial invasion of the amniotic cavity and amniotic fluid concentrations of prostaglandins and thromboxane B2 in term pregnancy. *Am J Obstet Gynecol* **168**, 1654-1664; discussion 1664-1658 (1993).

2. Lannon, S. M., Vanderhoeven, J. P., Eschenbach, D. A., Gravett, M. G. & Adams Waldorf, K. M. Synergy and interactions among biological pathways leading to preterm premature rupture of membranes. *Reproductive sciences* **21**, 1215-1227, doi:10.1177/1933719114534535 (2014).
3. Menon, R. & Fortunato, S. J. Infection and the role of inflammation in preterm premature rupture of the membranes. *Best Pract Res Clin Obstet Gynaecol* **21**, 467-478, doi:10.1016/j.bpobgyn.2007.01.008 (2007).
4. Goldenberg, R. L., Culhane, J. F., Iams, J. D. & Romero, R. Epidemiology and causes of preterm birth. *Lancet* **371**, 75-84, doi:10.1016/S0140-6736(08)60074-4 (2008).
5. Menon, R. & Richardson, L. S. Preterm prelabor rupture of the membranes: A disease of the fetal membranes. *Semin Perinatol* **41**, 409-419, doi:10.1053/j.semperi.2017.07.012 (2017).
6. Shim, S. S. *et al.* Clinical significance of intra-amniotic inflammation in patients with preterm premature rupture of membranes. *Am J Obstet Gynecol* **191**, 1339-1345, doi:10.1016/j.ajog.2004.06.085 (2004).
7. Romero, R. *et al.* Sterile and microbial-associated intra-amniotic inflammation in preterm prelabor rupture of membranes. *J Matern-Fetal Neo M* **28**, 1394-1409, doi:10.3109/14767058.2014.958463 (2015).
8. Waldorf, K. M. A. *et al.* Uterine overdistention induces preterm labor mediated by inflammation: observations in pregnant women and nonhuman primates. *American Journal of Obstetrics and Gynecology* **213**, doi:ARTN 830.e110.1016/j.ajog.2015.08.028 (2015).
9. Parry, S. & Strauss, J. F., 3rd. Premature rupture of the fetal membranes. *The New England journal of medicine* **338**, 663-670, doi:10.1056/NEJM199803053381006 (1998).
10. Wang, H. & Hirsch, E. Bacterially-induced preterm labor and regulation of prostaglandin-metabolizing enzyme expression in mice: the role of toll-like receptor 4. *Biol Reprod* **69**, 1957-1963, doi:10.1095/biolreprod.103.019620 (2003).
11. Grigsby, P. L., Poore, K. R., Hirst, J. J. & Jenkin, G. Inhibition of premature labor in sheep by a combined treatment of nimesulide, a prostaglandin synthase type 2 inhibitor, and atosiban, an oxytocin receptor antagonist. *Am J Obstet Gynecol* **183**, 649-657, doi:10.1067/mob.2000.106584 (2000).

2nd Editorial Decision

03 July 2018

Thank you for the submission of your revised manuscript to EMBO Molecular Medicine. We have now received the enclosed reports from the referees that were asked to re-assess it. As you will see the reviewers are now globally supportive and I am pleased to inform you that we will be able to accept your manuscript pending the following final amendments:

1) Please address the minor text changes commented by referee 3.

Please address the referee's comments in writing.

Please submit your revised manuscript within two weeks. I look forward to seeing a revised form of your manuscript as soon as possible.

***** Reviewer's comments *****

Referee #1 (Comments on Novelty/Model System for Author):

This is an important, well-conceived and well executed study.

Referee #1 (Remarks for Author):

The authors have now addressed all of my concerns.

Referee #3 (Remarks for Author):

Overall the authors did a great work in answering the queries of all three reviewers. There are some aspects that still need to be taken into consideration.

Again, the higher expression of ENaC found in placentas from women with spontaneous preterm labor as compared to women with term labor is not enough data to sustain the hypothesis on causality. The authors have now better dissected that this is not the case. However, at some passages it reads like this mechanism would exist in pre-term labour only (that is not the case)

The language needs to be revised, sentences like the following make the manuscript difficult to read "PROM is typically associated infection, although a number of literatures have indicated that infection is not always present in all or majority of PROM cases". This is just an example.

2nd Revision - authors' response

16 July 2018

***** Reviewer's comments *****

Referee #1 (Comments on Novelty/Model System for Author):

This is an important, well-conceived and well executed study.

Referee #1 (Remarks for Author):

The authors have now addressed all of my concerns.

Response: We thank the referee again for the positive comments and careful review of our manuscript.

Referee #3 (Remarks for Author):

Overall the authors did a great work in answering the queries of all three reviewers. There are some aspects that still need to be taken into consideration.

Again, the higher expression of ENaC found in placentas from women with spontaneous preterm labor as compared to women with term labor is not enough data to sustain the hypothesis on causality. The authors have now better dissected that this is not the case. However, at some passages it reads like this mechanism would exist in pre-term labour only (that is not the case)

Response: Thanks for the comment. We have carefully edited the related text. In particular, the passage (line 307-324) is adjusted to discuss the possible association of ENaC with preterm labor risk factors, but not the causality.

The language needs to be revised, sentences like the following make the manuscript difficult to read "PROM is typically associated infection, although a number of literatures have indicated that infection is not always present in all or majority of PROM cases". This is just an example.

Response: Thanks for the comment and spotting the error. The mentioned sentence has been language-corrected as "While PROM is typically associated with infection, a number of studies have shown that infection is not always present in PROM" (Line 341-343). The overall language has also been carefully re-edited.

YOU MUST COMPLETE ALL CELLS WITH A PINK BACKGROUND ↓
PLEASE NOTE THAT THIS CHECKLIST WILL BE PUBLISHED ALONGSIDE YOUR PAPER

Corresponding Author Name: Ye Chun RUAN
Journal Submitted to: EMBO Molecular Medicine
Manuscript Number: EMM-2018-08868-V2